# A selective and orally bioavailable VHL-recruiting PROTAC achieves SMARCA2 degradation in vivo

Targeted protein degradation offers an alternative modality to classical inhibition and holds the promise of addressing previously undruggable targets to provide novel therapeutic options for patients. Heterobifunctional molecules co-recruit a target protein and an E3 ligase, resulting in ubiquitylation and proteosome-dependent degradation of the target. In the clinic, the oral route of administration is the option of choice but has only been achieved so far by CRBN- recruiting bifunctional degrader molecules. We aimed to achieve orally bioavailable molecules that selectively degrade the BAF Chromatin Remodelling complex ATPase SMARCA2 over its closely related paralogue SMARCA4, to allow in vivo evaluation of the synthetic lethality concept of SMARCA2 dependency in SMARCA4-deficient cancers. Here we outline structure- and property-guided approaches that led to orally bioavailable VHL-recruiting degraders. Our tool compound, ACBI2, shows selective degradation of SMARCA2 over SMARCA4 in ex vivo human whole blood assays and in vivo efficacy in SMARCA4-deficient cancer models. This study demonstrates the feasibility for broadening the E3 ligase and physicochemical space that can be utilised for achieving oral efficacy with bifunctional molecules.

Heterobifunctional degrader molecules—also known as proteolysis-targeting chimeras (PROTACs)—target disease-causing proteins for destruction. They function by binding to both an E3 ligase and to the target protein. The induced proximity results in subsequent ubiquitination of the target protein, earmarking it for degradation by the proteasome. In cells, their mode of action enables degraders to achieve levels of target selectivity, breadth of target scope and efficacy not attainable with a classical inhibitor[1,2]. Due principally to the convenience of administration, oral dosing regimens dominate small molecule therapeutic delivery, however the design of orally available degraders is challenging because of their inflated physicochemical properties[3,4]. To date, the clinical translation of orally active degraders has been confined to the use of a single E3 ligase—Cereblon (CRBN), greatly limiting the potential therapeutic scope of PROTACs[5,6]. To the best of our knowledge, one VHL-recruiting PROTAC (DT2216) administered via intravenous infusion is currently in clinical trials and recent studies have demonstrated oral exposure for VHL PROTACs[7–9].

Here, we optimise VHL-based PROTACs to obtain quantifiable oral bioavailability in addition to in vivo efficacy. We also outline structure guided and hypothesis driven strategies broadly applicable towards the optimisation of large orally bioavailable bifunctional molecules. We found that design of linker composition and exit vector placement could be guided and rationalised by ternary co-crystal structures, yielding molecules which exhibit high potency and suitable pharmacokinetic properties to translate to oral in vivo efficacy. Our lead molecule ACBI2 is a full degrader of SMARCA2 and PBRM1, yet demonstrates strong selectivity over the highly similar paralogue SMARCA4 in human whole blood and shows consistent preferential degradation of SMARCA2 over SMARCA4 in all cell lines tested. This permits pharmacological evaluation of the synthetic lethality concept of selectively targeting SMARCA2 in SMARCA4-deficient cancers in vivo and in vitro[10–12]. We qualify ACBI2 as an orally bioavailable SMARCA2 degrader that will be made freely available to the community. We anticipate that our results and the methodologies employed

✉e-mail: harald.weinstabl@boehringer-ingelheim.com; w.farnaby@dundee.ac.uk

will provide a blueprint to arrive at oral efficacy of other E3-recruiting degraders.

## Results

### Discovery of a potent SMARCA2/4/PBRM1 binder to enable targeted protein degradation

PROTACs are often made by the conversion of existing protein of interest (POI) binders, allowing quick generation of protein degrader tool compounds. However, the optimisation of physicochemical properties to turn in vitro degrader tools into in vivo drugs often

remains challenging and can prohibit the progression of the compounds into the clinic. For the SMARCA2/4 bromodomains (BDs), two sub-micromolar binders have been described in the literature. PFI-3 was reported as a bona fide SMARCA2/4 BD-binding tool compound (Fig. 1a), albeit with modest affinity and a latent risk for unfavourable chemical stability[13]. In addition, Genentech reported a phenol-amino-pyridazine derivative (GEN-1) with attractive stability, physicochemical properties and affinity (Fig. 1a)[14]. We successfully used this motif to generate SMARCA2/4 degraders[15], but the polarity of the binding motif prohibited subsequent optimisation towards sufficient oral

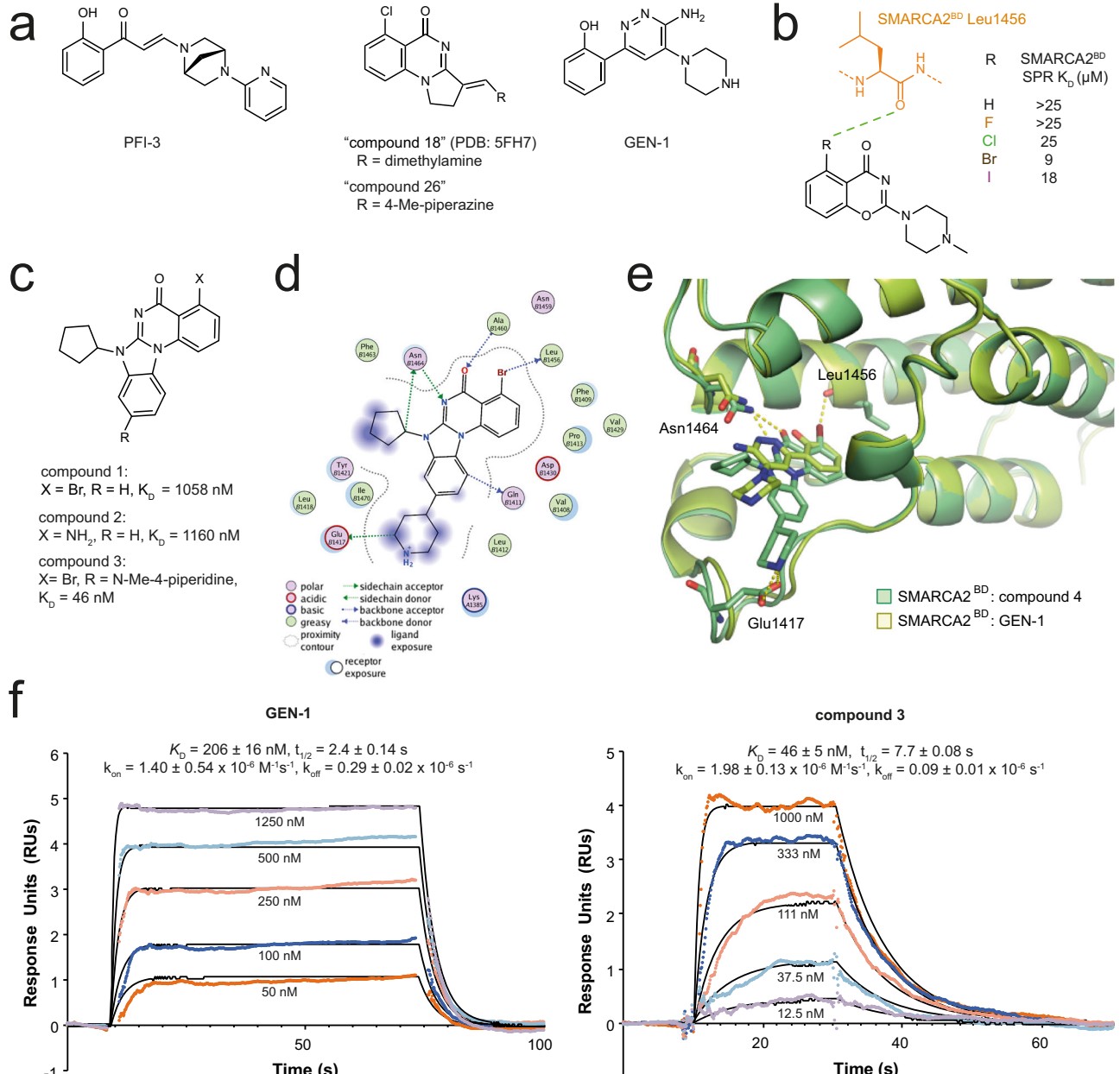

**Fig. 1 | Biophysical and structural characterisation of compound 3 and previously disclosed SMARCA bromodomain binders. a** Comparison of previously disclosed SMARCA binders with sub-μM binding affinity: PFI-3[13], GEN-1[14], "compound 18"[17] and "compound-26"[17]. **b** Optimisation of the halogen bonding interaction to Leu1456 based on an alternative benzoxazinone lead series (light brown, fluorine; green, chlorine; dark brown, bromine; purple, iodine; orange, SMARCA peptide. **c** SAR of SMARCA binding scaffold with reduced hydrogen bond donor count. **d** Binding mode and interactions of compound **4** (a close analogue of

compound **3**) with SMARCA2[BD] (PDB: 7Z78). See legend in figure for annotation of colours and symbols. **e** Superposition of the SMARCA2[BD]: compound **4** complex (green) with PDB: 6HAZ (yellow, GEN-1) highlighting key interactions towards Asn1464, Leu1456 and Glu1417. **f** SPR sensorgrams for binding of compound **3**/GEN-1 to SMARCA2[BD]; mean values reported with standard deviation (*n* = 3 independent experiments). Source data are provided as a Source Data file. Colours are different concentrations as indicated.

bioavailability for this VHL-based PROTAC. Driven by the multiple advantages of orally bioavailable drugs, such as a high patient acceptance, the possibility of non-sterile self-administration, cost-effectiveness and ease of large-scale manufacturing[16], we set out to obtain orally active SMARCA2 degraders.

As a first step, we elected to discover an alternative SMARCA2 bromodomain binder, incorporating only the absolute minimum of hydrogen bond donors, a high degree of rigidity and distinct and well-defined exit vectors. In addition, a reliable synthetic route, with the possibility for late-stage functionalisation, was considered essential. Inspired by Sutherell et al.[17], who previously identified binders for the bromodomain (BD) of PBRM1 (PBRM1[BD5]), which shares a high degree of similarity to the bromodomain of SMARCA2, we first characterised the molecular interactions that are mandatory for BD binding using a high-resolution crystal structure of "compound 18" (PDB: 5FH7, chemical structure in Fig. 1a). We found the halogen bond to the Met731[PBRM1] backbone (BB) carbonyl and the hydrogen bonding interaction between Asn739[PBRM1] and the quinazolinone core to be indispensable and witnessed SAR findings leading to "compound 26". Incorporating insights of halogen bond optimisation from an alternative benzoxazinone lead series (Fig. 1b) to the quinazoline core led to the design of scaffold compound 1 (Fig. 1c). We also evaluated the possibility to replace the halogen bonding interaction by a hydrogen bonding interaction in compound 2, however this was detrimental to binding affinity (Fig. 1c). Next, we hypothesised that placing a basic centre close to Glu1417[SMARCA2] could improve the binding affinity and would also balance the solubility of the compound. The attachment of a piperidine was superior to other linear and cyclic basic moieties, such as piperazines or amino-carbocycles, and led us to compound 3 as a purposefully designed SMARCA2/4 BD-binding motif for PROTAC generation (Fig. 1c). The expected binding mode of compound 3 was confirmed by solving a co-crystal structure of its nor-methyl analogue compound 4 with SMARCA2[BD] (Fig. 1d and Supplementary Fig. 1 PDB: 7Z78, 1.32 Å resolution, see Supplementary Table 1) revealing that the key interactions of the quinazoline core towards Asn1464 and Leu1456 as well as Glu1417 via the basic nitrogen of the piperidine are addressed in the SMARCA2[BD]. Overlay of compound 4 and GEN-1 (Fig. 1e) demonstrates the shift in binding mode from a bi-dentate (GEN-1) towards a mono-dentate interaction with Asn1464. Finally, SPR binding kinetics of compound 3 (SPR SMARCA2[BD] $K_D = 46$ nM) revealed a similar behaviour as previously reported for GEN-1 (SPR SMARCA2[BD] $K_D = 206$ nM), however with significantly reduced hydrogen bond donor count. (Fig. 1f, Supplementary Data 1), thus offering a superior starting point for the generation of orally bioavailable PROTACs.

**Identification of an in vivo active SMARCA2 degrader via exit vector hopping.** We previously reported potent dual degraders of SMARCA2/4, utilising a phenolic exit vector from the VHL ligand, and hypothesised that PROTACs based on our ligand compound 3 would offer additional opportunities for in vivo degrader optimisation[15]. PROTACs with PEG- and alkyl-based linkers showed moderate target degradation in RKO cells, with 27–75% maximal degradation ($D_{max}$) for SMARCA2 (Supplementary Data 2). Notably, compound 5 demonstrated partial degradation of SMARCA2 ($DC_{50} = 78$ nM, $D_{max} = 46\%$) while sparing SMARCA4 completely (Fig. 2a). Kinetic experiments demonstrated that compound 5 did not show degradation of SMARCA2 at 4 h, suggesting a slow rate of degradation (Supplementary Data 2). Rapid degradation kinetics may reduce the need for prolonged in vivo exposure. We have previously shown that E3 Ligase: PROTAC: POI ternary complex stability can impact the rate of degradation[18]. To support our understanding of the thermodynamics of ternary complex formation in this series, we established SPR and TR-FRET assays (Supplementary Data 1, 3), and solved a high-resolution co-crystal structure of the VCB: compound 5: SMARCA2[BD] complex (PDB: 7Z6L, Fig. 2b, see Supplementary Table 2). Consistent with poor target

degradation at 4 h, the data show moderate ternary complex binding affinities and a limited buried surface for this ternary complex (1837 Å$^2$). Whilst the position of SMARCA2[BD] may be influenced by crystal contacts formed in the closely packed crystal lattice (Supplementary Fig. 2a–g), the structure indicates a de novo protein-protein interaction (PPI) between Asn67 of VHL and Gln1469 of SMARCA2. It is noteworthy that SMARCA4 features a leucine residue in this position and is thus less capable of forming such a PPI, offering a possible rationale for the observed selectivity of compound 5. In pursuit of more stable complexes that could result in rapid degradation based on the co-crystal structure, we switched the VHL exit vector from the phenolic to the benzylic position (Fig. 2c), which was enabled by optimisation of synthetic methods (see Supplementary Information). This resulted in compound 6, a fast and potent dual degrader of SMARCA2/4 (SMARCA2 $DC_{50} = 2$ nM, $D_{max} = 77\%$; SMARCA4 $DC_{50} = 5$ nM, $D_{max} = 86\%$, in RKO cells after 4 h) (Fig. 2c, Supplementary Data 3) that displayed the expected anti-proliferative effect ($EC_{50} = 2$ nM, Supplementary Data 4) in a SMARCA4-deficient lung cancer cell line, NCI-H1568. Whole cell unbiased proteomics in this cell line showed compound 6 to be highly SMARCA2-selective, with concurrent degradation of PBRM1 the only other protein significantly degraded (Fig. 2d). We could rescue SMARCA2 degradation by inhibition of the VHL-HIF1-α interaction using VH298, neddylation inhibition using MLN4924 or proteasome inhibition by MG132 (Supplementary Fig. 3a–c). We solved the ternary complex crystal structure of VCB: compound 6: SMARCA2[BD] (PDB: 7Z77; Supplementary Fig. 2h, i; Supplementary Table 3), in which the compound adopts a different binding pose to that of compound 5 with an increased buried surface area of 2050 Å$^2$ (Fig. 2e), consistent with an observed increase in ternary complex half-life and ternary binding affinity in SPR and TR-FRET assays, compared with compound 5, offering an explanation for the differential cellular SAR (Fig. 2a, Supplementary Data 1, 3). PK studies revealed that the improved microsomal stability of compound 6 compared with compounds from the phenolic series (Supplementary Data 4) translated into a low clearance of 7 mL/min/kg in mouse, and the compound was quantitatively bioavailable upon subcutaneous administration (Supplementary Fig. 3d, Supplementary Data 5). Single dose subcutaneous treatment of an NCI-H1568 tumour xenograft model with compound 6 reduced median SMARCA2 levels in tumours by 90% relative to vehicle control-treated samples at 6 h after treatment, with slight recovery of the signal observed at 48 h after treatment, to a median decrease of 78% (Fig. 2f). This translated to a significant tumour growth inhibition (TGI) in two different treatment regimens that were both well tolerated (Fig. 2g, Supplementary Fig. 3e). In tumour samples collected at the end of the study, compound 6 treatment resulted in undetectable levels of SMARCA2 as assessed by IHC (Supplementary Fig. 3f, Supplementary Data 5). In summary, a structure-guided exit vector hop enabled discovery of small molecule degraders that form higher affinity complexes, enabling demonstration of prolonged biomarker modulation and anti-tumour efficacy in vivo in a SMARCA4-deficient xenograft model.

**A structure-guided approach to discover selective, orally bioavailable VHL PROTACs.** Findings in multiple protein degradation projects and recent publications have demonstrated that changing the linker even by only one atom can have a significant influence on molecular properties e.g. due to a different three-dimensional conformation of the molecule[19], but also selectivity by inducing different ternary complexes[20,21]. Guided by our finding that the short three-carbon linker used in compound 6 efficiently forms a high affinity ternary complex leading to potent degradation of SMARCA2/4, we elected to focus on a small set of alkyl- and ether-based analogues with the objective of improving selectivity and oral bioavailability. Linker elongation and branching led to compounds like compound 8 and compound 9, which show remarkable selectivity for SMARCA2 over

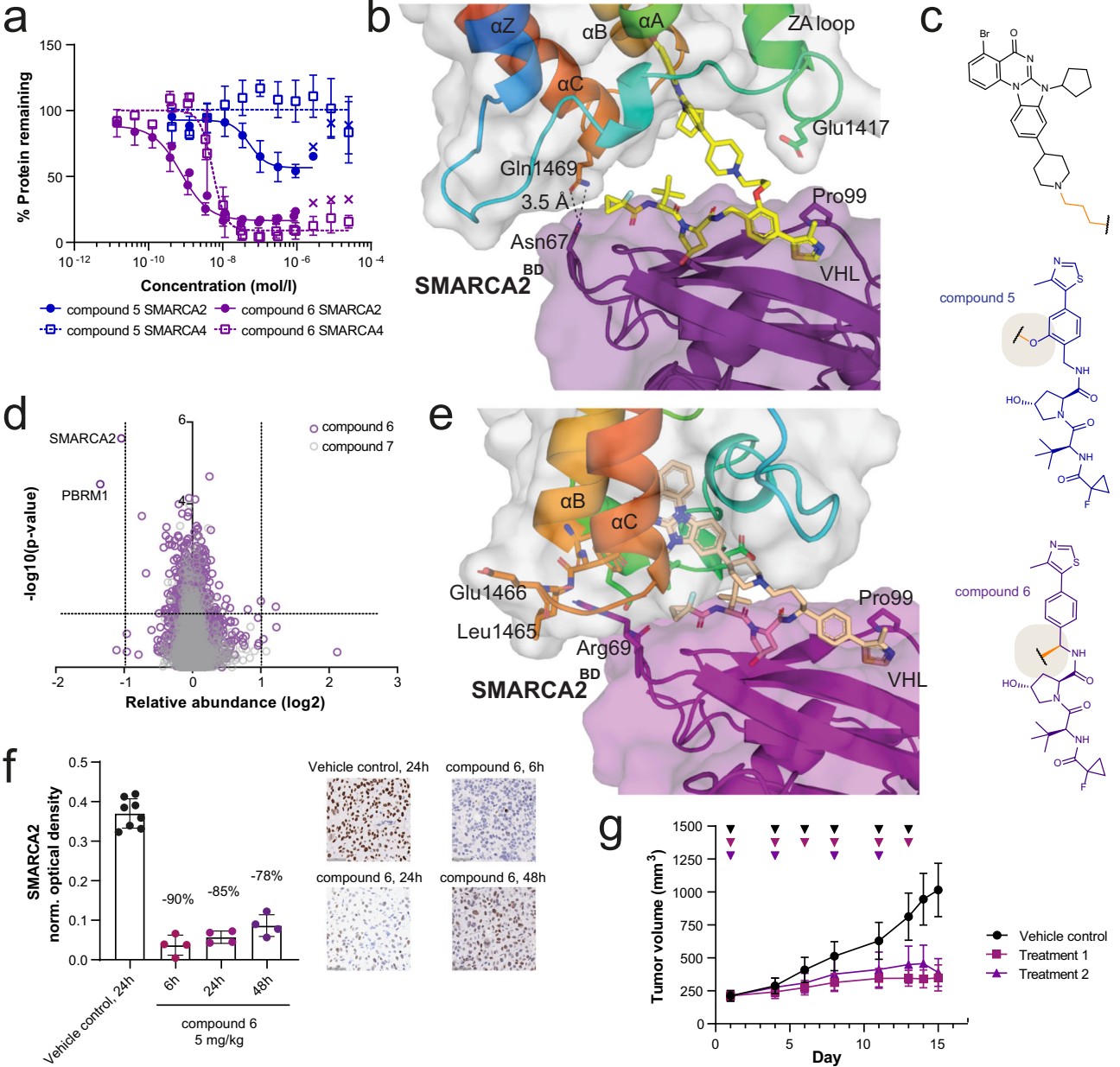

**Fig. 2 | Ternary structure-guided design of an in vivo active dual SMARCA2/4 degrader. a** SMARCA2 and SMARCA4 protein levels (continuous vs. dashed lines) measured by capillary electrophoresis normalised to DMSO control following compound treatment of RKO cells for 18 h (mean and standard deviation of $n = 6$ independent experiments for compound **6** SMARCA2, $n = 5$ for compound **6** SMARCA4 (both purple), $n = 3$ for compound **5** SMARCA2 and SMARCA4 (blue)). Values indicated by x are excluded from curve fit (hook effect). **b** Ternary X-ray crystal structure for VCB: compound **5**: SMARCA2BD (PDB: 7Z6L, purple = VHL, yellow = **5**, rainbow = SMARCA2BD). **c** Structures of compound **5** (blue) and compound **6** (purple) highlighting the VHL exit vector switch. **d** Effects of compound **6** (purple) and negative control compound **7** (grey, *cis*-hydroxyproline analogue of compound **6**, which abrogates binding to VHL) on the proteome of NCI-H1568 cells treated with the compounds at 100 nM for 4 h. Data are plotted as the $\log_2$ of the normalised fold change in abundance against $-\log_{10}$ of the $p$ value per protein from $n = 3$ independent experiments (two-tailed *t*-tests assuming equal variances). **e** Ternary X-ray crystal structure for VCB: compound **6**: SMARCA2BD (PDB: 7Z77, purple = VHL, pale orange = **6**, rainbow = SMARCA2BD). **f** NCI-H1568 tumour bearing mice (average tumour size ~260 mm³) were treated subcutaneously with 5 mg/kg compound **6** ($n = 4$ animals per time point) or vehicle control ($n = 8$ animals, 24 h only) and tumours were collected 6, 24 and 48 h after treatment (shades of purple). SMARCA2 levels were determined by IHC staining (representative images are shown, scale bar = 50 µm). Each datapoint represents the background-normalised optical density (OD) within the viable tumour area of one tumour section, corresponding to one individual tumour. Mean OD and standard deviations are indicated. Percentages represent the median levels of SMARCA2 signal decrease relative to vehicle (black). **g** NCI-H1568 tumour bearing mice (average tumour size ~210 mm³) were treated subcutaneously with 5 mg/kg compound **6** in two treatment schedules (Treatment 1/2, see methods for details, square vs. triangle in shades of purple), resulting in a TGI of 77 and 84%, respectively, at day 15 after the start of treatment (adj. $p$ value = 0.0002 for either regimen vs. control (black), one-tailed U-test with Bonferroni-Holm correction, mean and standard deviation of 10 animals per group). Triangles indicate days of treatment. Source data are provided as a Source Data file.

SMARCA4 degradation (Supplementary Data 2; Fig. 3a). To gain a better understanding for the molecular basis of this selectivity, we again turned to crystal structure analysis. We were able to solve the ternary crystal structure of compound **10** (PDB: 7Z76; Supplementary Table 4), a close analogue of compound **9**, that only differs in the VHL binder site where the fluorine is replaced by a dimethyl amino group (Fig. 3a, b). This ternary structure revealed that an extensive network of de novo electrostatic interactions between SMARCA2 and VHL was

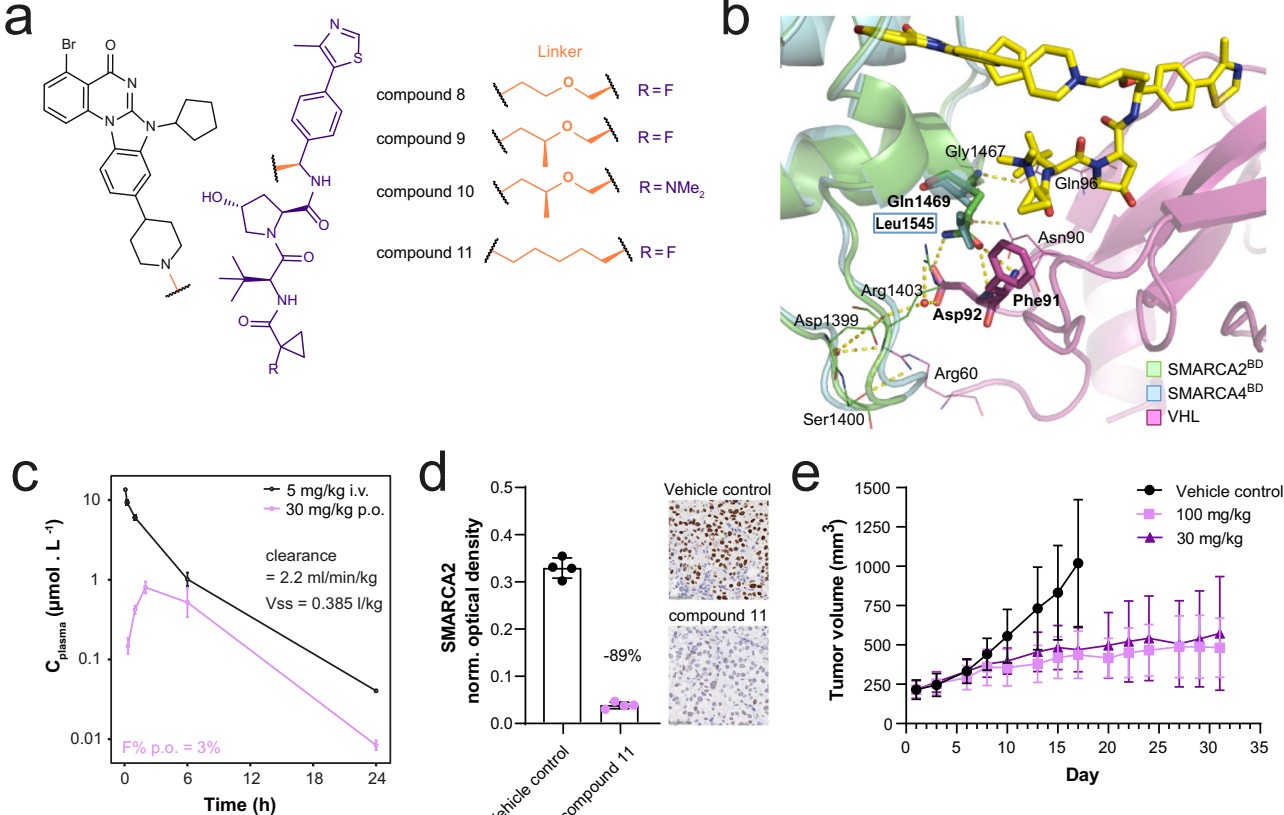

**Fig. 3 | Design and in vivo evaluation of an orally bioavailable VHL-based PROTAC. a** Structures of compound **8**, compound **9**, compound **10** and compound **11**. SMARCA2 binding scaffold in black. VHL binding scaffold in purple. Exit vector and linker motifs connecting SMARCA2 and VHL binding scaffolds in orange. **b** Co-crystal structure of VCB: compound **10**: SMARCA2$^{BD}$ in ribbon representation (PDB: 7Z76). VCB is shown in magenta, SMARCA2$^{BD}$ is shown in green with bound PROTAC shown in sticks with yellow carbons overlayed with SMARCA4$^{BD}$ in ribbon representation shown in blue. Represented are the key PPIs between VCB and SMARCA2$^{BD}$/SMARCA4$^{BD}$, highlighting the selectivity-inducing hydrogen bonding between Gln1469 of SMARCA2$^{BD}$ and VCB vs. Leu1545 in SMARCA4$^{BD}$. **c** Plasma profiles of compound **11** in mouse after administration of 5 mg/kg i.v. (black) or 30 mg/kg p.o. (pink). Displayed are mean and standard deviation of $n = 3$ animals. **d** NCI-H1568 tumour bearing mice (average tumour size ~290 mm³) were treated orally with 100 mg/kg compound **11** (pink) or vehicle control (black; $n = 4$ animals per group) and tumours were collected 48 h after treatment. SMARCA2 levels were determined by IHC staining (representative images are shown, scale bar = 50 μm). Each datapoint represents the background-normalised OD within the viable tumour area of one tumour section, corresponding to one individual tumour. Mean OD levels and standard deviations are indicated. Percentages represent the median level of SMARCA2 signal decrease relative to vehicle control-treated samples. **e** NCI-H1568 tumour bearing mice (average tumour size ~220 mm³) were treated orally with compound **11** at 30 or 100 mg/kg daily (square vs. triangle in shades of pink). At day 17 after treatment start, TGI was 76% for the 100 mg/kg and 64% for the 30 mg/kg treatment group (adj. *p* value = 0.0009 for either regimen vs. control (black), one-tailed U-test with Bonferroni-Holm correction). Values represent the mean of 10 animals per group, error bars indicate standard deviation. Source data are provided as a Source Data file.

formed, leading to the formation of a ternary complex significantly different from those previously observed (Supplementary Fig. 4a–c). Furthermore, the SMARCA2-specific residue Gln1469 was involved in VHL: SMARCA2$^{BD}$ interactions, as previously observed for the SMARCA2-selective molecule compound **5**, albeit in this case in the context of a different overall arrangement of the two proteins. In SMARCA2, Gln1469 positively interacts with VHL residues Phe91 and Asp92 via hydrogen bonds, an interaction that cannot occur in SMARCA4 that harbours Leu1545 instead of Gln1469. In summary, linker elongation with an oxygen and linker branching gave rise to compounds that were selective towards SMARCA2, a preference rationalised by ternary complex structures. However, despite good microsomal stability and measurable Caco-2 permeability (Supplementary Data 4), the compounds showed a high efflux ratio, preventing oral bioavailability.

Learning that linker elongation and branching improves selectivity, we hypothesised that this should also apply to the more lipophilic all-carbon series. Linker elongation from three to five carbon atoms also resulted in a slight SMARCA2/4 selectivity improvement, by two- to

threefold within the all-carbon linker series, as observed by comparing compound **6** with compound **11** (Supplementary Data 2). Furthermore, linker elongation led to an improved permeability for compound **11**, which, due to its good microsomal stability and moderate solubility (Supplementary Data 4), constituted an orally bioavailable SMARCA2 VHL PROTAC, despite a high efflux ratio (Fig. 3c, Supplementary Data 5). We tested compound **11** in an NCI-H1568 xenograft study, treating mice orally with 100 mg/kg, and evaluated SMARCA2 levels in viable tumour tissue 48 h after treatment. A median decrease of 89% compared to vehicle control-treated tumours was detected (Fig. 3d). In subsequent studies, mice were treated orally with 30 mg/kg or 100 mg/kg compound **11** daily. A significant TGI of 64% within the 30 mg/kg and 76% within the 100 mg/kg treatment group could be reached at day 17 of the experiment (Fig. 3e). The compound was well tolerated in both dose groups as assessed by body weight changes (Supplementary Fig. 4d). At the end of the study, SMARCA2 levels were undetectable by IHC in most treated tumour samples from both treatment groups (Supplementary Fig. 4e). Together, compound **11** achieved significant in vivo activity and quantifiable oral bioavailability.

**ACBI2 is an orally active degrader that selectively degrades SMARCA2 over SMARCA4.** Due to the differences of the respective E3 ligase binders, VHL PROTACs tend to have 2D physicochemical properties further from classical oral druggable space as compared with CRBN PROTACs[3,4]. Nevertheless, it has been shown that in some cases PROTACs can adopt more compact 3D conformations that yield 3D polar surface and radius of gyration more consistent with that required for permeability[19,20]. Reduction of the polar surface area by switching from the ether linkage as shown for compounds **8–10** to an all-carbon linkage, led to an improved permeability and reduced efflux ratio resulting in acceptable oral bioavailability for compound **11** (Supplementary Data 4). In addition, we hypothesised that if more compact conformations of compound **11** could be enabled via changes to the alkyl linker, we could further enhance absorption, reduce efflux, and ultimately improve oral bioavailability. We therefore incorporated an additional methyl group on the apolar C5 linker of compound **11**. This modification yielded ACBI2, a highly potent VHL PROTAC (EC$_{50}$ = 7 nM), which degrades SMARCA2 with a > 30-fold window over SMARCA4 in RKO cells (SMARCA2 DC$_{50}$ = 1 nM, SMARCA4 DC$_{50}$ = 32 nM) (Supplementary Data 2). The branched linker reduced ACBI2 efflux, which directly resulted in an improved oral bioavailability of 22% (Fig. 4a, Supplementary Data 4).

Molecular dynamics (MD) simulations and NMR studies comparing compound **11** and ACBI2 were performed to understand the impacts of linker branching on conformational restraint. Conformational ensembles from simulations indeed showed a trend towards collapsed structures with lower radius of gyration leading to lower free energies for ACBI2 (Supplementary Data 6). Consequently, lower polar surface area values tended to be favoured within the conformational ensemble of ACBI2 (Supplementary Fig. 5a). NMR studies were carried out to evaluate long-range nuclear Overhauser effects (NOEs) (Fig. 4b). We determined a long-range NOE from the t-butyl group to the piperidine group for ACBI2 that was not detectable in compound **11**. Furthermore, under identical experimental conditions, the sign of the NOE crosspeaks was different for the two compounds in CDCl$_3$-d, indicating a different degree of mobility and therefore compactness of the compounds, with ACBI2 having the more compact structure (see Supplementary Figs. 7–11; Supplementary Tables 5, 6). Taken with measurements in Caco-2 cells, in which **11** and ACBI2 show similar passive permeabilities in the presence of an efflux inhibitor, these data suggest the more compact structure of ACBI2 led to a reduction in the efflux ratio, contributing to improved oral bioavailability (Supplementary Data 4).

Encouraged by the improved oral bioavailability and selectivity profile of ACBI2, we characterised the compound in more detail in vitro. A panel of cell lines showed varying levels of sensitivity to ACBI2, correlating with genetic dependency on SMARCA2 due to mutation or lower expression of SMARCA4 (Fig. 4c, Supplementary Fig. 5b). Accordingly, ACBI2 treatment caused rapid and complete degradation of SMARCA2 in two sensitive cell lines (A549 and NCI-H1568; Supplementary Fig. 5c). As for compound **6**, we could rescue SMARCA2 (and PBRM1) degradation by inhibition of VHL, neddylation or the proteasome (Supplementary Fig. 5d, e). We also detected decreased mRNA levels of *KRT80*, a gene that is transcriptionally regulated downstream of SMARCA2 and has been proposed as a biomarker associated with SMARCA2 inhibition[22] (Supplementary Fig. 5f), confirming functional perturbation of BAF complex roles by ACBI2. Unbiased whole cell proteomic analysis demonstrated proteome-wide selectivity for degradation of SMARCA2 and, as expected from the POI binding moiety, PBRM1 in protein lysates prepared from NCI-H1568 cells, similar to compound **6** (Figs. 4d, 2d). Interestingly, we observed that the extent of selectivity of SMARCA2 degradation over SMARCA4 varied in cell lines expressing both proteins (HCT116 and RKO, Supplementary Fig. 6a). We investigated if this might be correlated with differences in half-life or re-synthesis rates of either SMARCA2 or

SMARCA4, and indeed observed a trend towards higher selectivity in RKO, the cell line with shorter half-lives and faster re-synthesis of both SMARCA2 and SMARCA4 (Supplementary Figs. 6b, 5c). We tested ACBI2 selectivity in five additional cell lines and confirmed preferential degradation of SMARCA2 over SMARCA4 in all of them (Supplementary Fig. 6d). We cannot formally rule out other differences between these cell lines as contributors to differential selectivity (e.g. proliferation rate, mutations or expression levels of SMARCA2, SMARCA4 and VHL or ratios thereof), but did not observe obvious trends towards either of those in this small cell line panel.

We went on to test ACBI2 in vivo and observed dose-dependent SMARCA2 degradation in NCI-H1568 and A549 engrafted tumour bearing mice following short-term treatment (Fig. 4e, f). Correspondingly, ACBI2 (administered at 80 mg/kg orally once daily) significantly inhibited tumour growth in an A549 xenograft model (Fig. 4g) and was well tolerated (Supplementary Fig. 6e). SMARCA2 protein levels in most compound-treated tumours collected at the end of this study were decreased to background levels (Fig. 4h). Finally, we tested ACBI2 ex vivo treatment of human whole blood, obtained from three different healthy donors and observed significant degradation of SMARCA2 with clear selectivity over SMARCA4 (Fig. 4i). Together, these data demonstrate that oral bioavailability in combination with preferential degradation of one close paralog, SMARCA2, over the other, SMARCA4, can be achieved in vitro and in vivo with our VHL-based protein degrader ACBI2.

## Discussion

An oral route of administration for a new small molecule therapeutic is currently considered the rule rather than the exception. Despite an increasing number of orally dosed bifunctional degraders in the clinic, all those disclosed to date rely on the CRBN E3 ligase recognition subunit[23–25]. Whilst successful, this restriction greatly limits the long-term therapeutic scope and is predicated to some degree on an assumption that larger E3 ligase recruiting motifs cannot yield orally available PROTACs[26]. Here, we introduce three principles for arriving at orally available VHL-based degraders that we believe to be of general utility: Firstly, the de novo design of structurally different and potent protein of interest binders that display improved physicochemical properties at the outset of bifunctional degrader design. Secondly, crystallographic knowledge of ternary complex binding modes guiding exploration of additional exit vector space to achieve more stable complexes and consequently more potent and faster degraders. Lastly, small linker modifications influencing compound conformations leading to more compact arrangements with reduced 3D polar surface area and radius of gyration. As has been shown previously in the field of targeted protein degradation[27,28], we were also able to identify compounds that discriminate and preferentially degrade highly homologous target proteins (here, the bromodomains of SMARCA2 over SMARCA4) without appreciable differences in binding affinity for the target ligand alone. The degree to which selectivity was achieved was highly dependent on the linker employed. We observed higher selectivity for compounds from the ether series (e.g. compound **10**) compared with ACBI2. Structurally, these compounds differ both in composition (e.g. ether vs. all carbon) and length. Both biochemical and ternary co-crystal structures illuminate possible contributory factors for enhanced selectivity of the ether series. For example, ether series compounds demonstrate greater differences in cooperativity between SMARCA2 and SMARCA4 compared with ACBI2 (Supplementary Data 3), and compound **10** features a key ternary complex binding interaction with SMARCA2-specific residue Gln1469 (Fig. 3b, PDB: 7Z76). Given this and the difference in linker composition we would hypothesise the likelihood of a different ternary complex arrangement being formed for ACBI2 is high, contributing to its lower SMARCA2 selectivity compared with ether series molecules, though this was accepted as a trade-off due to the improved oral bioavailability

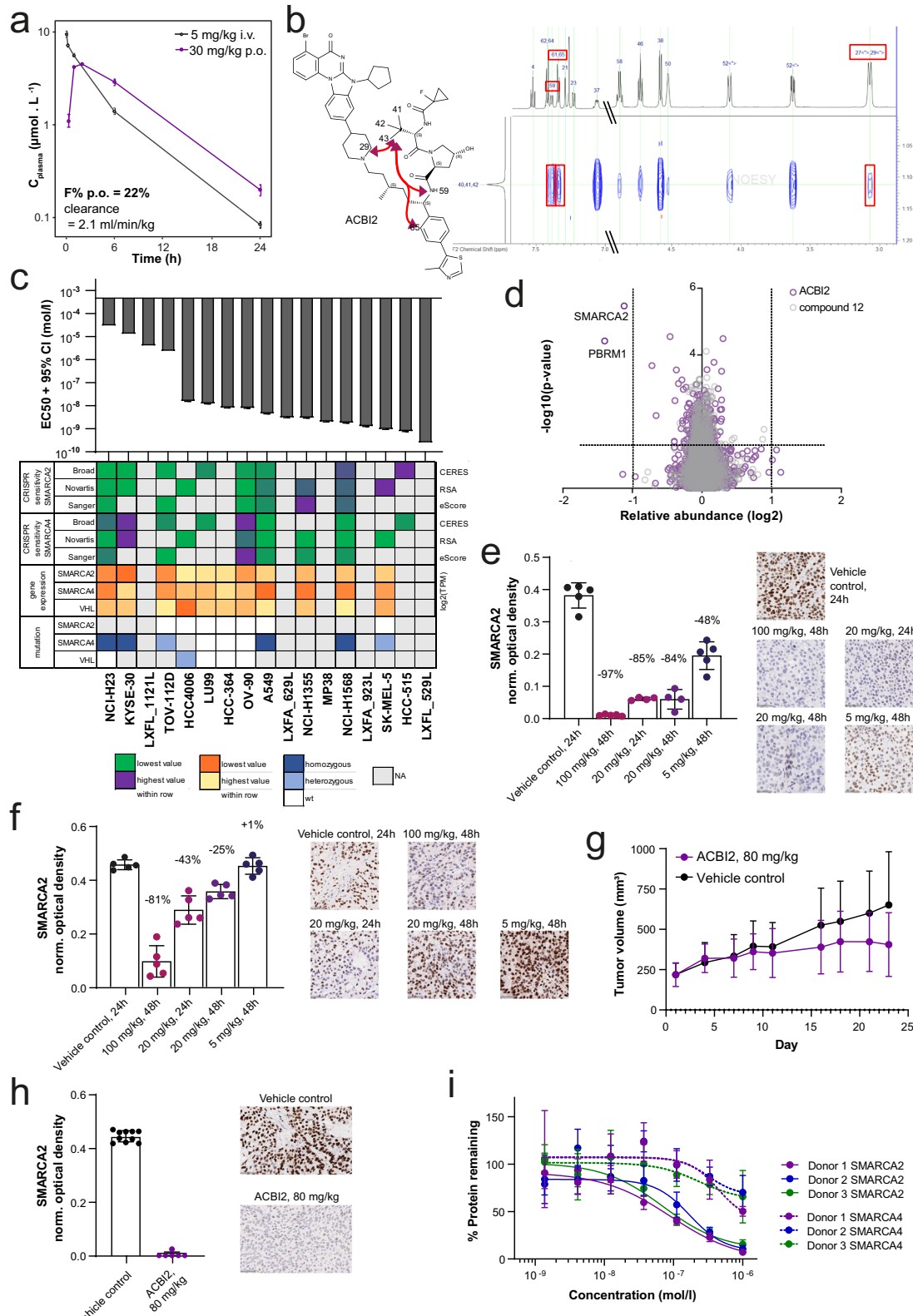

of ACBI2. Unfortunately, we were not able to solve a ternary co-crystal structure to ascertain ACBI2's ternary binding mode.

BAF (SWI/SNF) chromatin remodelling complexes play critical roles in cancer[29,30]. For example, it has recently been shown that androgen receptor (AR) and forkhead box A1 (FOXA1) expressing prostate cancer cells are sensitive to simultaneous degradation of BAF complex subunits SMARCA2, SMARCA4 and PBRM1[31]. The synthetic lethality between SMARCA2 and SMARCA4, resulting in

sensitivity of SMARCA4-deficient cells to loss of SMARCA2, has been discovered and validated by genetic methods[10–12], but pharmacological validation and exploitation of this synthetic lethal relationship has been hampered by the lack of suitably selective small molecules, in particular for effective in vivo use in animal models[15,22]. Here we show that ACBI2 is capable of inducing near-complete degradation of SMARCA2 in mouse lung cancer xenograft models that leads to tumour growth inhibition. Cellular studies demonstrate degradation

**Fig. 4 | ACBI2 is an orally bioavailable degrader that preferentially degrades SMARCA2 and induces lung cancer tumour growth inhibition. a** Plasma profiles of ACBI2 in mouse after administration of 5 mg/kg i.v. (black) or 30 mg/kg p.o. (purple). Displayed are mean and standard deviation of $n = 3$ animals. **b** Structure of ACBI2 (black) and selected long-range NOEs in 2D NOESY spectra (red). **c** The indicated cell lines were treated with ACBI2 for 144–192 h, and cell viability was measured using CellTiter Glo ($n = 2$ independent experiments for A549, HCC-364, HCC-515, HCC4006, KYSE-30, LU99, LXFA_629L, LXFA_923L, LXFL_529L, LXFL_1121L, MP38, NCI-H23, NCI-H1355, OV-90, SK-MEL-5, TOV-112D; $n = 7$ for NCI-H1568). Displayed are $EC_{50}$ values with 95% confidence interval from 4-parametric logistic curve fit. Heatmap provides sensitivity to genetic depletion by CRISPR, gene expression and mutation data from DepMap/CCLE, colours are scaled for each row, i.e. across cell lines, but separate for each parameter according to legend. **d** Effects of ACBI2 (purple) and negative control compound **12** (grey, *cis*-hydroxyproline analogue of ACBI2 which is not capable of binding VHL) on the proteome of NCI-H1568 cells treated with the compounds at 100 nM for 4 h. Data are plotted as the $\log_2$ of the normalised fold change in abundance against $-\log_{10}$ of the $p$ value per protein from $n = 3$ independent experiments (two-tailed $t$-tests assuming equal variances). **e** NCI-H1568 (average tumour size at treatment start ~470 mm³) or (**f**) A549 (average tumour size at treatment start ~360 mm³) tumour bearing mice were treated orally with 100 mg/kg, 20 mg/kg, 5 mg/kg ACBI2 or vehicle control ($n = 5$ animals per group; shades of purple) and tumours collected 24 or 48 h after treatment. Tumours from one animal (NCI-H1568 20 mg/kg ACBI2 24 and 48 h) could not be analysed due to necrosis. SMARCA2 levels were determined by IHC staining (representative images are shown, scale bar = 50 μm). Each datapoint represents the background-normalised OD within the viable tumour area of one tumour section, corresponding to an individual tumour. Mean OD levels and standard deviations are indicated. Percentages represent median levels of SMARCA2 signal decrease relative to vehicle (black). **g** A549 tumour bearing mice (average tumour size ~220 mm³) were treated orally with 80 mg/kg ACBI2 once daily (purple). At day 21 of treatment, a TGI of 47% ($p$ value = 0.0351 vs. control (black), one-tailed U-test) was measured (mean and standard deviation of 10 animals per group). **h** At the end of the study in (**g**), tumours were collected and SMARCA2 levels determined as for (**e**, **f**); representative images are shown, scale bar = 50 μm. Mean OD levels and standard deviations are indicated in the graphs. The median level of SMARCA2 staining in ACBI2 treated tumours was reduced to background levels. **i** Human whole blood from three healthy donors (purple, blue, green) was treated with the indicated concentrations of ACBI2 for 18 h in the dark. Protein was extracted from PBMCs and relative SMARCA2 and SMARCA4 levels (each normalised to GAPDH, continuous vs. dashed line) measured using capillary electrophoresis (mean and standard deviation from three biological replicates, displayed relative to DMSO control). Source data are provided as a Source Data file.

of PBRM1, the contribution of PBRM1 depletion to phenotypes can therefore not be excluded. However, it has previously been shown with ACBI1, a potent degrader of SMARCA2, SMARCA4 and PBRM1, that the anti-proliferative effects in NCI-H1568 (SMARCA4 deficient) cells can be negated by overexpression of SMARCA2 but not PBRM1[15]. At the same time, ACBI2 offers a clear window of selectivity between SMARCA2 and SMARCA4 degradation in human whole blood and a consistent preference for the degradation of SMARCA2 over SMARCA4 in cell lines expressing both ATPases. Nevertheless, it is notable that despite efficient degradation of SMARCA2 in vivo, only tumour stasis was observed upon compound treatment in the models studied herein. This is unexpected given the strong effects observed upon SMARCA2 deletion or knockdown in functional genomic studies[10–12], suggesting a disconnect with pharmacological degradation. While it cannot entirely be excluded that more efficacious degraders may cause stronger effects, it is also possible that cells can more readily adapt to loss of SMARCA2 and SMARCA4 in vivo than in vitro, highlighting the need for in vivo validation of therapeutic concepts. The possibility that other indications such as prostate cancer or multiple myeloma are more dependent on dual loss of SMARCA2 and SMARCA4 remains[31]. In either case, appropriate drug combinations could enhance in vivo efficacy and warrant dedicated investigation in the future. To promote further understanding in the community, ACBI2 will be made freely available upon request via the opnMe innovation platform (https://opnme.com/molecules/smarca2-acbi2). We anticipate that our study will aid thinking around the design of orally efficacious bifunctional molecules and hope that the studies described here will encourage others to explore the chemical and biological space that may be utilised to discover orally active bifunctional small molecule therapeutics.

## Methods
### Ethics approval
The authors confirm that the research in this study complies with all relevant ethical regulations.

All animal studies were approved by the internal ethics committee (called "ethics committee") of Boehringer Ingelheim RCV GmbH & Co KG in the department of Cancer Pharmacology and Disease Positioning. Furthermore, all protocols were approved by the Austrian governmental committee (MA 60 Veterinary office; approval numbers GZ: 154399/2018/16 and GZ: MA 58-670393-2019-18).

Human whole blood for ex vivo degradation assays was purchased from the Austrian Red Cross, who always obtain samples under informed consent in accordance with relevant guidelines, regulations, and internal approvals to ensure ethics and informed consent of donors.

### Chemical synthesis
A list of final compounds is compiled in Supplementary Fig. 12. Full details of synthetic procedures (including schemes: Supplementary Figs. 13–21) and NMR spectra of final compounds (Supplementary Figs. 32–63) are provided as Supplementary Methods.

### Protein crystallography and protein production
Protein production for SMARCA2 and the VCB complex was done as previously described[15]:

Wild type and mutant versions of human proteins were used for all protein expression, as follows: VHL (UniProt accession number: P40337), ElonginC (Q15369), ElonginB (Q15370) and the bromodomains (BDs) of SMARCA2 (SMARCA2BD; P51531-2, residues 1373-1493 with additional N-terminal SM residues (cloning artefact)), SMARCA4 (SMARCA4BD; P51532, residues 1448–1569 with additional N-terminal SM residues (cloning artefact)), and the fifth bromodomain of PBRM1 (PBRM1BD5, Q86U86, residues 645–766). SMARCA4BD was provided by the Structural Genomics Consortium (SGC), Toronto[32] and SMARCA2BD and PRM1BD5 were synthesised by GeneArt then subcloned into pDEST15 vectors (Invitrogen). The VCB complex was expressed and purified as described previously, with the modification that 0.3 mM IPTG was used for induction of expression[27]. Briefly, N terminally His$_6$ tagged VHL (54–213), ElonginC (17–112) and ElonginB (1–104) were co expressed and the complex isolated by Ni affinity chromatography, the His$_6$ tag was removed using TEV protease, and the complex further purified by anion exchange and size exclusion chromatography (SEC). The VCB$^{R69A}$ mutant, in which R69 of VHL (54–213) was mutated to alanine, was generated using a QuickChange II site directed mutagenesis kit (Agilent) according to the manufacturer's instructions and expressed and purified as for VBC. Both were stored in 20 mM 4-(2-hydroxyethyl)−1-piperazineethanesulfonic acid (HEPES), 150 mM sodium chloride and 1 mM tris(2-carboxyethyl)phosphine (TCEP) pH 7.5. SMARCA2BD, SMARCA4BD and PBRM1BD5 were expressed in *Escherichia coli* BL21(DE3) as N-terminal GST fusion proteins with a TEV protease cleavage site. Expression and purification of these proteins has been described previously[32,33]. Starter cultures were grown overnight at 37 °C in 10 mL of Luria-Bertani (LB) medium with ampicillin (100 μg/mL). The starter cultures were diluted (1:100) in Terrific Broth (TB) medium with ampicillin (100 μg/mL) and grown in a shaking incubator at 37 °C to an optical density ($OD_{600}$) ~1 before the temperature was lowered for induction (to 23 °C for SMARCA2BD or 18 °C for SMARCA4BD and

PRRM1[BD5]). Expression was induced using IPTG (final concentration 0.3 mM for SMARCA2[BD], PBRM1[BD5], 0.4 mM for SMARCA4[BD]) for 16 h at the specified temperatures. Cells were harvested by centrifugation and stored at −80 °C prior to purification. Cells were resuspended in Lysis buffer and lysed by sonication (20 pulses of 5 s for 4 min) using a Sonopuls HD 3080 (Bandelin, Berlin, Germany) or by homogenisation using a Stansted Pressure Cell Homogeniser (Stansted Fluid Power). The Lysis buffers were as follows: for SMARCA2[BD]: 50 mM HEPES, 500 mM sodium chloride, 5% glycerol, 5 mM dithiothreitol (DTT); for SMARCA4[BD]: 25 mM HEPES, 0.3 M sodium chloride, 5% glycerol, 10 mM DTT, pH 7.5; for PBRM1[BD5]: 25 mM HEPES, 300 mM sodium chloride, 5% glycerol, 10 mM DTT, pH 7.8; in each case supplemented with complete protease inhibitors (Roche). Affinity purification was performed using Glutathione Sepharose 4B (GE Healthcare) in batch mode or on-column. Cleavage of the GST-tag was performed using TEV protease for 16 h at 4 °C, either on column, or in solution following elution of the GST-tagged BDs with Lysis buffer containing 20 mM reduced L glutathione (Sigma Aldrich). For SMARCA2[BD], prior to TEV protease cleavage the eluted GST-tagged BD was first dialysed into desalting buffer (20 mM HEPES, 250 mM sodium chloride, pH 7.0 + 0.5% glycerol). Where TEV cleavage was performed in solution, a second affinity (GST-trap) column purification step was carried out to remove the GST-tag and uncleaved GST-tagged protein. Proteins were further purified by SEC (HiLoad Superdex-75, 16/600) (GE Healthcare) and stored in Storage buffer (for SMARCA2[BD]: 20 mM HEPES, 150 mM sodium chloride, pH 7.5; for SMARCA4[BD]: 10 mM HEPES, 0.5 M NaCl, 5% glycerol, pH 7.5; for PBRM1[BD5]: 20 mM HEPES, 300 mM NaCl, 5 mM DTT, pH 7.8). For AlphaScreen assays, eluted GST-tagged BDs were purified directly by SEC in the respective storage buffers without TEV cleavage. All chromatography purification steps were performed either at room temperature or 4 °C using an ÄKTA FPLC purification system (GE Healthcare) or a plastic Econo-Pac column (Bio-Rad).

Protein crystallisation was performed using a vapour diffusion method with 96-well sitting drop plates. For the binary SMARCA2[BD]: compound **4** complex, SMARCA2[BD] apo crystals were soaked overnight with a 2 mM compound **4** DMSO stock solution. Apo crystals were generated by mixing 200 nL of SMARCA2 protein with an equal volume of reservoir solution consisting of 8% ethylene glycol, 25% PEG 6000, 0.1 M HEPES pH 8.0 and 10 mM zinc chloride. The compound **4**: SMARCA2[BD] complex was refined to Rwork and Rfree values of 18.7 and 21.0%, respectively, with 99.4% of the residues in Ramachandran favoured regions as validated with MOLPROBITY.

For the VCB: compound **5**: SMARCA2[BD] ternary complex, VCB, PROTAC and SMARCA2[BD] were mixed in a 1:1:1 stoichiometric ratio in 20 mM HEPES, pH 7.5, 150 mM sodium chloride, 1 mM TCEP, 2% DMSO, incubated for 5 min at room temperature and concentrated to a final concentration of approximately 10 mg/ml. Drops were prepared by mixing 1 μl of the ternary complex with 1 μl of well solution and crystallised at 4 °C using the hanging-drop vapour diffusion method. Crystals were obtained in 33% (v/v) glycerol ethoxylate, 0.2 M ammonium chloride, 0.1 M HEPES, pH 7.5. Harvested crystals were flash cooled in liquid nitrogen following gradual equilibration into cryoprotectant solution consisting of 25% (v/v) ethylene glycol in 35% (v/v) glycerol ethoxylate, 0.2 M ammonium chloride, 0.1 M HEPES, pH 7.5. Diffraction data were collected at Diamond Light Source beamline I24 (λ = 0.9686 Å) using a Pilatus3 6 M detector and processed using XDS[34]. The crystals belonged to space group P 21 with unit cell parameters a = 47.3, b = 86.8, c = 59.3 Å and α = 90°, β = 98.9°, γ = 90° and contained one copy of the ternary complex per asymmetric unit. The structure was solved by molecular replacement using PHASER[35] with VCB coordinates derived from the VCB: MZ1: Brd4BD2 complex (PDB: 5T35) and SMARCA2[BD] (PDB: 4QY4) as search models. Subsequent iterative model building and refinement was done according to standard protocols using CCP4[36], COOT[37], and autoBUSTER (Global Phasing Ltd). The structure was refined to Rwork and Rfree values of 19.6

and 23.5%, respectively, with 98.0% of the residues in Ramachandran favoured regions as validated with MOLPROBITY[38]. Buried surface areas were calculated with PISA[39].

For the remaining ternary complexes, SMARCA2, VCB complex and the PROTAC molecule were mixed in a 1:1:1 molar ratio, concentrated to ~8 mg/mL in a buffer containing 20 mM HEPES pH 7.5, 100 mM NaCl, 1 mM TCEP and incubated with equal amounts of reservoir solution. For compound **6**, the reservoir solution was 20% PEG 3350, 0.1 M BIS-TRIS propane pH 7.5 and 200 mM sodium formate. For compound **10**, the solution consisted of 14.5% PEG 3350, 0.1 M BIS-TRIS propane pH 5.8 and 100 mM sodium iodide. For data collection, crystals were frozen in liquid nitrogen with the respective reservoir solution with 20–25% ethylene glycol added. Synchrotron data was collected at the Swiss Light Source (Villigen) and processed with autoPROC[40] using STARANISO[41] for the determination of resolution limits. Model building was done in iterative cycles using COOT[37] and autoBUSTER[42]. The structure of the VCB: compound **6**: SMARCA2[BD] complex was refined to Rwork and Rfree values of 19.3 and 25.1%, respectively, with 97.1% of the residues in Ramachandran favoured regions as validated with MOLPROBITY. The VCB: compound **10**: SMARCA2[BD] complex was refined to Rwork and Rfree values of 17.8 and 20.2%, respectively, with 98.4% of the residues in Ramachandran favoured regions as validated with MOLPROBITY.

## SPR experiments
SPR data were acquired on Biacore 8K or T200 instruments (Cytiva). Target proteins were immobilised at 25 °C on CM5 chips by amine coupling (EDC/NHS, Cytiva) in HBS-P+ running buffer (pH 7.4), containing 2 mM TCEP. After surface activation with EDC/NHS (contact time 420 s, flow rate 10 μl/min) SMARCA2[BD] and SMARCA4[BD] at 0.01–0.05 mg/ml in coupling buffer (10 mM Na-Acetate pH 6.5, 0.005% TWEEN 20 and 50 μM PFI-3[13]) were immobilised to a density of 100–5000 Response Units (RU). The reference surface was subsequently deactivated using 1 M ethanolamine. For VCB immobilisation, streptavidin (Sigma Aldrich, prepared at 1 mg/ml in 10 mM sodium acetate coupling buffer, pH 5.0) was immobilised by amine coupling to a density of 3000–5000 RU after which biotinylated VCB complex (0.125 mM in running buffer) was streptavidin-coupled to a density of 100–500 RU. Biotinylated VCB was prepared as previously described[27]: in brief, the complex was mixed 1:1 with EZ-Link NHS–PEG4–biotin (Thermo Scientific) and incubated at room temperature for 1 h. Unreacted NHS-biotin was removed using a PD-10 desalting column (Cytiva) into 20 mM HEPES, pH 7.5, 150 mM sodium chloride and 1 mM DTT. The reference surface was generated by deactivating the EDC/NHS-treated surface with 1 M ethanolamine. All interaction experiments were done at 6 °C in running buffer (20 mM TRIS, pH 8.3, 150 mM potassium chloride, 2 mM magnesium chloride, 2 mM TCEP, 0.005% TWEEN 20, 1% dimethyl sulfoxide). For ternary complex measurements, a sensor chip surface with VCB immobilised was used (preparation as described above). Experiments were run in dual-inject mode with 10 μM SMARCA present during the injection and dissociation phase. Sensorgrams from reference surfaces and blank injections were subtracted from the raw data prior to data analysis using Biacore Insight software. Affinity and binding kinetic parameters were determined by global fitting using the 1:1 interaction model with a term for mass-transport included.

## Cell culture
Cell lines were typically cultured in flasks (75 cm²) at sub-confluency, were free of mycoplasma contamination in regular checks, authenticated by STR profiling (Eurofins Genomics) and kept at low passage numbers in humidified incubators at 37 °C and 5% CO₂. Cell lines were obtained from The University of Texas Southwestern Medical Centre (HCC-364 RRID:CVCL_5134, cat.# NA; HCC-515 RRID:CVCL_5136, cat.# NA), DSMZ (KYSE-30 RRID:CVCL_1351, cat.# ACC 351; Caco-2

RRID:CVCL_0025, cat.# ACC 169), JCRB (LU99 RRID:CVCL_3015, cat.# JCRB0080), Charles River (LXFA 629L RRID:CVCL_D189, cat.# NA; LXFA 923L, cat.# NA; LXFL 529L RRID:CVCL_D085, cat.# NA; LXFL 1121L, cat.# NA;) and ATCC (all others). The following media were used: RPMI 1640 (ATCC) with Glutamax + 10% FCS + sodium pyruvate + 10 mM HEPES + 0.25% glucose (HCC4006 RRID:CVCL_1269, cat.# CRL-2871; HCC-364; HCT-15 RRID:CVCL_0292, cat.# CCL-225; LU99; NCI-H1568 RRID:CVCL_1476, cat.# CRL-5876; NCI-H23 RRID:CVCL_1547, cat.# CRL-5800), RPMI-1640 (Anprotec) with glutamine and HEPES (AC-LM-0054) + 10% FCS + 50 µg/ml gentamicin (LXFA 629L, LXFA 923L, LXFL 529L, LXFL 1121L), DMEM + 10% FCS (RKO RRID:CVCL_0504, cat.# CRL-2577; Caco-2), McCoy's + 10% FCS (HCT116 RRID:CVCL_0291, cat.# CCL-247), F12-K + 10% FCS (A549 RRID:CVCL_0023, cat.# CCL-185), ACL-4 + 5% FCS (NCI-H1355 RRID:CVCL_1464, cat.# CRL-5865), EMEM with Glutamax + 10% FCS sodium pyruvate (SK-MEL-5 RRID:CVCL_0527, cat.# HTB-70; RKO), 1:1 RPMI: F12 Ham's + 10% FCS (KYSE-30), 1:1 MCDB105 + 1.5 g/l sodium bicarbonate: M199 + 2.2 g/l sodium bicarbonate + 15% FCS (OV-90 RRID:CVCL_3768, cat.# CRL-11732; TOV-112D RRID:CVCL_3612, cat.# CRL-11731), DMEM + 10% FCS + sodium pyruvate + NEAA (SK-N-AS RRID:CVCL_1700, cat.# CRL-2137), RPMI 1640 + 5% FCS (HCC-515), RPMI 1640 + 20% FCS (MP38 RRID:CVCL_4D11, cat.# CRL-3296). VH298 was purchased from Tocris, MG132 and MLN4294 were purchased from Merck.

Cell line annotation (CRISPR depletion, gene expression and mutation data) was obtained from Novartis' and the Broad and Sanger Institutes' Cancer Dependency Map (DepMap) and Cancer Cell Line Encyclopaedia (CCLE) projects (https://depmap.org/portal/).

### Cell viability assays
500–2000 cells were seeded in white, clear bottom 96-well plates (Corning or Perkin Elmer) in 180–200 µl medium per well. The next day, compounds were added using a digital dispenser and a T0 sample was measured for reference. Cells were incubated for 144–192 h, and viability (luminescence) was measured using CellTiter Glo or CellTiter Glo 2.0 (Promega) according to manufacturer's instructions after equilibrating cells and reagents at room temperature and 10–20 min incubation time while shaking. Values were displayed relative to negative controls (DMSO) and curves were fitted using a 4-parametric logistic model.

### SMARCA2/4 capillary electrophoresis protein assays
130,000–200,000 cells were seeded in 24-well plates and incubated until settled down or overnight. Compounds were dissolved in DMSO and added to cells in indicated concentrations using a digital dispenser following an optional media change. Cells were incubated for 4 or 18 h as indicated. For SMARCA2/4 half-life determination, 150,000 or 200,000 cells were seeded in 24-well plates. The next day, 50 µg/ml cycloheximide (Sigma) was added and cells were incubated for 1–24 h. For SMARCA2/4 re-synthesis measurements, 200,000 cells were seeded in 24-well plates. The next day, 100 nM compound **6** was added for 4 h to degrade SMARCA2/4, and a sample was taken as baseline for normalisation. To allow re-synthesis, a VHL binder (compound **32**, 25 or 50 µM) was then added for 3 or 24 h. Medium was removed, cells were washed with PBS and lysed with 80 µl Lysis Buffer (MSD Tris Lysis Buffer (#R60TX-2) + Halt Inhibitor Cocktail (100x) + Benzonase 0.5 µL/ml). Samples were frozen at −80 °C and thawed at room temperature before use. Samples were transferred to a V-bottom plate and insoluble debris was pelleted by centrifugation for 5 min at maximum speed. The supernatant was transferred to a fresh plate. Master Mix and Ladder were prepared according to manufacturer's instructions for 12–230 kDa Wes Separation Module, 8 × 25 capillary cartridges, Protein Simple #SM-W004 with Anti-Rabbit Detection Module for Wes, Peggy Sue or Sally Sue, Protein Simple #DM-001. 4.8 µl lysate were mixed with 1.2 µl Master Mix to achieve a protein concentration of ~0.5 µg/µl.

Antibodies were diluted in Antibody Diluent II (SMARCA2 Sigma #HPA029981 RRID:AB_10602406 1:25–100, SMARCA4 Cell Signalling #49360 clone D1Q7F RRID:AB_2728743 1:15–120 or Abcam #ab110641 clone EPNCIR111A RRID:AB_10861578 1:60, GAPDH Abcam #ab9485 RRID:AB_307275 1:250). Wes plates were prepared and run according to manufacturer's instructions. Proteins were quantified with accompanying Compass Software. SMARCA2/4 protein levels were normalised to GAPDH. Values were displayed relative to negative controls (DMSO) unless indicated otherwise. Where applicable, curves were fitted using a 4-parametric logistic model.

### qPCR
350,000–500,000 cells were seeded in 6-well plates and compounds were added the next day at the indicated concentrations after pre-dilution in medium. Cells were incubated for 18 h, harvested, and RNA was extracted using the RNeasy Mini Kit (Qiagen) according to manufacturer's instructions. A DNAse digestion step was included. 1 µg RNA was used for cDNA synthesis with the SuperScript VILO cDNA synthesis kit (Invitrogen) according to manufacturer's instructions. TAQMAN qPCR was then performed using the QuantiTect Multiplex PCR kit (Qiagen) with probes for KRT80 (Hs01372365_m1, Applied Biosystems) and human GAPDH as endogenous control (4326317E, VIC/MGB probe, Primer Limited) for normalisation in a duplex assay over 45 cycles with 100 ng cDNA input. Relative mRNA levels were calculated using the ΔΔCt method and displayed relative to the DMSO control. Curves were fitted using a 4-parametric logistic model.

### Imaging-based SMARCA2/4 protein assay in RKO cells
1250 cells per well in 60 µl medium were plated in flat bottom, poly-lysin coated 384-well plates (CellCarrier Ultra, Perkin Elmer). The next day, test compounds (10 µl of serial dilutions) and DMSO controls were diluted in DMEM medium such that the final DMSO content was <1% or were added using an ultrasound dispensing system. 4 wells were reserved for a background measurement. Cells were incubated for 24 h and then fixed by adding 25 µl 7.4% formaldehyde (0.2% Triton-X-100) in PBS for 15 min at room temperature. After aspirating the fixing solution, the cells were washed once with 25 µl PBS. 25 µl of blocking buffer (10% goat serum in PBS) was added and cells were incubated for 30 min, then washed once with PBS. Cells were stained with 20 µl of SMARCA2/4 primary antibody solution (Sigma #HPA029981 RRID:AB_10602406 1:1000 or Cell Signalling #52251 clone E9O6 RRID:AB_2799410 1:1000 in PBS with 10% FCS) for 2–4 h at room temperature. Cells were again washed once with PBS. 25 µl 5 µg/ml Hoechst 33342 (1:2000, Invitrogen H1399) was added for detection of nuclei, together with Alexa Fluor 647 goat anti mouse IgG (Invitrogen #A32728 RRID:AB_2633277 1:1000) or Alexa Fluor 488 goat anti-rabbit IgG (Invitrogen #A11034 RRID:AB_25762171:1000) in PBS with 10% FCS, and cells were incubated for 60 min at room temperature. The cell layer was then washed once with 25 µl PBS, the wells were filled with 25 µl PBS and the plates were sealed with an adhesive sheet. They were then imaged on the Opera Phoenix (mean intensity in the nucleus). Results were computed as percent of controls ((value of test compound − background)/(value of the negative control DMSO−background) multiplied by 100). EC$_{50}$ values were computed using a 4-parametric logistic model.

### Pharmacokinetic analyses
Compound concentrations in plasma aliquots were measured by quantitative HPLC-MS/MS using an internal standard. Calibration and quality control samples were prepared using blank plasma from untreated animals. Samples were precipitated with acetonitrile and injected into a HPLC system (Agilent 1200). Separation was performed by gradients of 5 mmol/L ammonium acetate pH 4.0 and acetonitrile with 0.1% formic acid on a 2.1 mm by 50 mm XBridge BEH C18 reversed-phase column with 2.5 µm particles (Waters). The HPLC was interfaced

by ESI operated in positive ionisation mode to a triple quadrupole mass spectrometer (5000 or 6500+ Triple Quad System, SCIEX) operated in multiple reaction monitoring mode. Transitions were 532.4 to 432.8 m/z for BI01810284, 525.6 to 425.7 m/z for BI01802983 and 1021.6 to 624.2 m/z for BI01580883. Chromatograms were analyzed with Analyst (SCIEX) and pharmacokinetic parameters were calculated by non-compartmental analysis using BI-proprietary software.

## Solubility testing

Compound solubility was determined by dilution of a 10 mmol/l compound solution in DMSO into buffer to a final concentration of 125 µg/ml. Dilution into a 1:1 mixture of acetonitrile and water was used as reference. After 24 h, the incubations were filtrated, and the filtrate was analyzed by LC-UV.

## Microsomal stability

The degradation kinetics of 1 µmol/l compound in 0.5 mg/ml liver microsomes were inferred in 100 mM Tris-HCl pH 7.5, 6.5 mM MgCl$_2$ and 1 mM NADPH at 37 °C. Reactions were terminated by addition of acetonitrile and precipitates separated by centrifugation. Compound concentrations in supernatants were measured by HPLC-MS/MS and clearance was calculated from compound half-lives using the well-stirred liver model.

## Plasma protein binding

Binding of compound to plasma proteins was determined by equilibrium dialysis of 3 µmol/L compound in plasma (or plasma dilutions in PBS) against PBS through an 8 kDa molecular-weight cut-off cellulose membrane (RED device, Thermo Fisher) at 37 °C for 5 h. After incubation, aliquots from donor and acceptor compartments were precipitated and the concentrations in the supernatants were determined by quantitative LC-MS/MS. Calibration and quality control samples were prepared using blank plasma and internal standard. The fraction unbound was calculated as ratio of the compound concentration in the acceptor compartment to the concentration in the donor compartment.

## Bidirectional permeability measurement in Caco-2 cells

Bidirectional permeability of test compounds across a Caco-2 cell monolayer was measured as described[43] with a modification of pre-incubation time[44]: Briefly, Caco-2 cells were seeded onto Transwell inserts (#3379, Corning, Wiesbaden, Germany) at a density of 160000 cells/cm$^2$ and cultured in DMEM (high glucose) containing 10% fetal bovine serum for 14–21 days. Cells were incubated with culture media containing 1 µM test compound for 24 h. After the preincubation period, culture media were removed and fresh transport buffer (128.13 mM NaCl, 5.36 mM KCl, 1 mM MgSO$_4$, 1.8 mM CaCl$_2$, 4.17 mM NaHCO$_3$, 1.19 mM Na$_2$HPO$_4$, 0.41 mM NaH$_2$PO$_4$, 15 mM 2-[4-(2-hydroxyethyl)piperazin-1-yl]ethanesulfonic acid (HEPES), 20 mM glucose, pH 7.4, 0.25% bovine serum albumin) containing 1 µM test compound was added to the apical (apical to basal) or basal (basal to apical) compartment (donor compartment), transport buffer without test compound was added to the opposite compartment (receiver compartment). Samples were taken at different time points for up to 2 h. Test compound in the samples was quantified with LC-MS/MS. To elucidate the role of drug transporter P-gp in the transcellular transport, permeability measurement was performed in the absence and presence of the selective P-gp inhibitor zosuquidar (5 µM final concentration). Apparent permeability coefficients (Papp,AB, Papp,BA) were calculated as follows:

$$Papp,AB = Q\_AB/((C0 \times s \times t)) \qquad (1)$$

$$Papp,BA = Q\_BA/((C0 \times s \times t)) \qquad (2)$$

where Q is the amount of compound recovered in the receiver compartment after the incubation time t, C0 the initial compound concentration given to the donor compartment, and s the surface area of the Transwell inserts. Efflux ratio is calculated as the quotient of Papp,BA (mean of duplicate) to Papp,AB (mean of duplicate). The P-gp substrate apafant and one low permeable compound (BI internal reference, Papp ≈ 3 × 10$^{-7}$ cm/s, no efflux) were included in every assay plate. In addition, Transepithelial electrical resistance (TEER) values were measured for each plate before the permeability assay. All three parameters (efflux of the reference substrates, Papp values of the low permeable compound, and TEER values) were used to ensure the quality of the assays.

## Animals and xenograft experiments

Female BomTac:NMRI-*Foxn1$^{nu}$* mice were obtained from Taconic Denmark at an age of 6–8 weeks. After arrival at the local AAALAC-accredited animal facility at Boehringer Ingelheim RCV GmbH & Co KG mice were allowed to adjust to housing conditions for at least 5 days before the start of the experiment, i.e. mice in all experiments were 7–9 weeks old. Mice were group-housed under pathogen-free and controlled environmental conditions (21 ± 1.5 °C temperature, 55 ± 10% humidity) and handled according to the institutional, governmental and European Union guidelines (Austrian Animal Protection Laws, GV-SOLAS and FELASA. Animal studies were approved as described in the Ethics section. Food and water were provided ad libitum.

To establish subcutaneous tumours, mice were injected with 5 × 10$^6$ NCI-H1568 in PBS with 5% FCS or with 1 × 10$^7$ A549 cells in PBS with 5% FCS. Tumour diameters were measured with a caliper three times a week. The volume of each tumour (in mm$^3$) was calculated according to the formula [tumour volume = length × diameter$^2$ × π/6]. To monitor side effects of treatment, mice were inspected daily for abnormalities and body weight was determined daily after the start of treatment. Animals were sacrificed when the tumours reached a size of 1500 mm$^3$. Mice were randomised into treatment groups when the average tumour size reached ~210–220 mm$^3$. Group sizes were calculated individually for each tumour model based on tumour growth during model establishment experiments. A power analysis was performed using a sample size calculator (https://www.stat.ubc.ca/~rollin/stats/ssize/n2.html). For both models used in the study, 10 mice per group were used for each experiment. All administrations were dosed with 10 ml/kg (s.c. and oral). Control mice were dosed subcutaneously with 10% HP-β-CD in 50% Ringer solution and orally with 15% HP-β-CD, that means the control mouse treatment was corresponding to the solvent of the compounds. Compound **6** was dosed subcutaneously either in a d1-4 q3d, d6-8 q2d, d11-13 q2d treatment (Treatment 1) or a q3 or 4d treatment (Treatment 2). Compound **11** was dosed at 30 or 100 mg/kg and ACBI2 at 80 mg/kg, both compounds orally with a daily dosing.

For the evaluation of the statistical significance of tumour growth inhibition, a one-tailed nonparametric Mann-Whitney-Wilcoxon U-test was performed, based on the hypothesis that an effect would only be measurable in one direction (i.e. expectation of tumour inhibition but not tumour stimulation). Analysis was performed on the day indicated for each experiment. The *p* values obtained from the U-test were adjusted using the Bonferroni-Holm correction. By convention, *p* values ≤0.05 indicate significance of differences.

## Immunohistochemistry and imaging analysis

Xenograft samples were fixed in 10% formaldehyde for 24 h and later moved to ethanol and embedded in paraffin. 2 µm-thick sections were cut using a microtome, then placed on glass slides (KLINIPATH, silan printer slides PR-S:001), and subsequently dewaxed. SMARCA2 immunohistochemistry was carried out on the Leica BOND RX platform (Leica Biosystems) according to manufacturer's instructions using a human-specific SMARCA2 antibody (Cell Signalling #11966

clone D9E8B RRID:AB_2797783 1:400) where the tissue underwent a heat-induced epitope retrieval for 20 min. After staining, the slides were cover-slipped with Shandon Consul-Mount glass covers, scanned using a slide scanner (3D HISTECH Ltd.). All slides were reviewed and evaluated for quality by a board-certified MD specialist in Anatomic Pathology (PC). Imaging analysis was performed using the digital pathology platform HALO (Indica Labs). A tissue-classifying algorithm was trained to selectively recognise viable tumour tissue against stroma, necrosis, and skin. The tissue classification output for each scan was reviewed and manually edited as necessary. A cell detection and scoring algorithm was trained to measure DAB optical density (OD) in the nuclei of tumour cells. A positivity threshold for DAB OD was determined by normalisation with respect to the DAB OD as calculated from bona fide negative tissue (e.g. murine stroma as background). The average, background-normalised DAB OD of tumour cell nuclei was used to quantitate SMARCA2 expression in each xenograft sample.

### Ex vivo human whole blood assay

The assay was performed on human whole blood from three healthy donors, purchased from the Austrian Red Cross who always obtain samples under informed consent. Aliquots of whole blood were treated in triplicate with a gradient of ACBI2 (starting at 1 μM and six 1:3 dilutions). Samples were pivoted at room temperature for 18 h in the dark. Subsequently, peripheral blood mononucleated cells (PBMCs) were isolated from individual aliquots using SepMate Tubes (StemCell) and Lymphoprep (StemCell) medium according to the manufacturer's instructions and protein lysates were prepared using MSD Lysis buffer (Mesocale Discovery) supplemented with 1:100 Halt Phosphatase-Protease Inhibitor Cocktail (Thermo Fisher), 0.5 μl/ml benzonase (Novagen) and 10 mM DTT. SMARCA2 and SMARCA4 protein levels were determined using capillary electrophoresis (Bio-Techne) according to manufacturer's instructions. Incubation with primary antibodies for SMARCA2 (Cell Signalling #11966 clone D9E8B RRID:AB_2797783 1:25), SMARCA4 (Abcam #ab110641 clone EPNCIR111A RRID:AB_10861578 1:25) and GAPDH (Abcam, #ab9485 RRID:AB_307275 1:1000) were performed for 30 min.

### Proteomics

Cells were seeded at $5 \times 10^6$ cells on a 100 mm plate 24 h before treatment. Cells were treated in triplicate by addition of test compounds at 100 nM. After 4 h, the cells were washed twice with 10 ml cold PBS and lysed in 500 μL of 100 mM TEAB with 5% (w/v) SDS. The lysate was pulse sonicated briefly and then centrifuged at $15,000 \times g$ for 10 min. Samples were quantified using a micro-BCA protein assay kit (Thermo Fisher Scientific). 300 μg of each sample was reduced with DTT, alkalised with iodoacetamide and double-digested with trypsin using the modified S-TRAP mini (ProtiFi) protocol. Peptide quantification was done using Pierce™ Quantitative Fluorometric Peptide Assay and equal amount from each sample was labelled using TMTpro™ 16plex Label Reagent Set (Thermo Fisher Scientific) as per the manufacturer's instructions. The samples were then pooled and desalted using a 7 mm, 3 mL C18 SPE cartridge column (Empore, 3 M). The pooled and desalted sample was fractionated using high pH reverse-phase chromatography on an XBridge peptide BEH column (130 Å, 3.5 μm, 2.1 × 150 mm, Waters) on an Ultimate 3000 HPLC system (Thermo Scientific/Dionex). Buffers A (10 mM ammonium formate in water, pH 9) and B (10 mM ammonium formate in 90% acetonitrile, pH 9) were used over a linear gradient of 2% to 100% buffer B over 80 min at a flow rate of 200 μL/min. 80 fractions were collected using a WPS-3000 FC auto-sampler (Thermo Scientific) before concatenation into 20 fractions based on the UV signal of each fraction. All the fractions were dried in a Genevac EZ-2 concentrator and resuspended in 1% formic acid for MS analysis. The fractions were analysed sequentially on a Q Exactive HF Hybrid Quadrupole-Orbitrap Mass Spectrometer

(Thermo Scientific) coupled to an Dionex Ultimate 3000 RS (Thermo Scientific). Buffers A (0.1% formic acid in water) and B (0.1% formic acid in 80% acetonitrile) were used over a linear gradient from 5% to 35% buffer B over 125 min and then from 35% buffer B to 98% buffer B in 2 min at a constant flow rate of 300 nl/min. The column temperature was 50 °C. The mass spectrometer was operated in data dependent mode with a single MS survey scan from 335 to 1600 $m/z$ followed by 15 sequential $m/z$ dependent MS2 scans. The 15 most intense precursor ions were sequentially fragmented by higher energy collision dissociation (HCD). The MS1 isolation window was set to 0.7 $m/z$ and the resolution set at 120,000. MS2 resolution was set at 60,000. The AGC targets for MS1 and MS2 were set at $3 \times 10^6$ ions and $1 \times 10^5$ ions, respectively. The normalised collision energy was set at 32%. The maximum ion injection times for MS1 and MS2 were set at 50 ms and 200 ms, respectively. The mass accuracy was checked before the initiation of sample analysis. The raw MS data files for all 20 fractions were merged and searched against the Uniprot-sprot-Human-Canonical database (FASTA file released in November 2019) by Maxquant software 2.0.3.0 for protein identification and TMT reporter ion quantitation. The Maxquant parameters were set as follows: enzyme used Trypsin/P; maximum number of missed cleavages equal to two; precursor mass tolerance equal to 10 p.p.m.; fragment mass tolerance equal to 20 p.p.m.; variable modifications: oxidation (M), dioxidation (MW), acetyl (N-term), deamidation (NQ), Gln -> pyro-Glu (Q N-term); fixed modifications: carbamidomethyl (C). Peptide and protein data was filtered by applying a 1% false discovery rate followed by exclusion of proteins with less than two unique peptides. Quantified proteins were filtered if the absolute fold-change difference between the three DMSO replicates was ≥1.5.

### Western blot

Cells were seeded into 6-well plates 24 h before treatment. Next day, several wells were treated with compounds or DMSO as indicated, washed with PBS and lysed with lysis buffer (1% Triton X-100, 150 mM NaCl, 1 mM EDTA, 50 mM Tris pH 7.4, protease inhibitor cocktail (Roche), 50 units/mL benzonase nuclease (Sigma)). Lysates were cleared by centrifugation at 4 °C, at $15,800 \times g$ for 10 min and the supernatants stored at −20 °C. Protein concentration was determined by BCA assay (Pierce) and the absorbance at 562 nm measured by spectrophotometry (NanoDrop ND1000). Samples were separated by SDS-PAGE using 20 μg of protein per well of NuPAGE Novex 4–12% BIS-TRIS gels (Invitrogen) and transferred to 0.2 μm pore nitrocellulose membrane (Amersham) by wet transfer. Western blot images were obtained through detection of SMARCA2 (Sigma #HPA029981 RRID:AB_10602406 1:1000), SMARCA4 (Abcam #ab108318 clone EPR3912 RRID:AB_10889900 1:1000), PBRM1 (Bethyl Laboratories #A301-591A RRID:AB_1078808 1:1000), β-actin (Cell Signalling #4970 clone 13E5 RRID:AB_2223172 1:2500) and GAPDH (Abcam #ab9485 RRID:AB_307275 1:2500) antibodies with IRDye 800CW donkey anti-rabbit secondary antibody (LI-COR #926-32213 RRID:AB_621848 1:10000) using a ChemiDoc MP imaging system (Bio-Rad). Western blots were quantified using Image Studio Lite (Licor, version 5.2) with normalisation to loading control and DMSO and further analysed using GraphPad Prism (version 9.2.0).

### Molecular dynamics simulations

1 μs accelerated molecular dynamics (aMD) simulations were performed in AMBER[45] to exhaustively sample the conformational space of PROTACs compound **11** and ACBI2 in explicit aqueous solution in analogy to earlier established protocols for sampling of peptidic macrocycles[46]. Reweighted conformational ensembles (first and last frame from aMD sampling are provided as Supplementary Data 6) were used to calculate the potential of mean force along the molecular descriptors radius of gyration as a measure of compactness as well as polar surface area to characterise exposed polar regions. Trajectory

slicing into five splits of 200 ns demonstrated coverage of conformational space and yielded standard deviations for free energy profiles.

## Reporting summary

Further information on research design is available in the Nature Research Reporting Summary linked to this article.

## Data availability

Proteomics data generated during the study are available via ProteomeXchange Consortium via the PRIDE[45] partner repository, under the dataset identifier PXD032239 [https://www.ebi.ac.uk/pride/archive/projects/pxd032239] (Effects on the proteome of NCI-H1568 cells for compound 6 and ACBI2). X-ray co-crystal data have been deposited to the PDB under accession codes 7Z78 (compound 4 in complex with SMARCA2[BD]), 7Z6L (VCB: compound **5**: SMARCA2[BD] complex), 7Z77 (VCB: compound **6**: SMARCA2[BD] complex), and 7Z76 (VCB: compound **10** SMARCA2[BD] complex). [1]H and [13]C NMR spectra for PROTACs are provided in the Supplementary Information. All other data generated for all Tables, Figures and Supplementary Figures are available in the Supplementary Data files. The coordinate files of the first and last frame from aMD sampling are provided as Supplementary Data 6. Plasmids generated in this study are available from the corresponding authors upon request due to restrictions in plasmid repositories for non-academic researchers. Source data are provided with this paper.

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

## Acknowledgements
The authors would like to thank Erich Spielvogel, Patrick Werni, Teresa Gmaschitz, Karin-Stefanie Hofbauer, Martina Kohla, Oliver Petermann, Johannes Wachter, Carina Riedmüller, Daniel Zach, Sabine Kallenda, Christian Salamon, Janine Rippka, Markus Graf-Gabriel, Teresa Puchner, Sonja Porits, Katrin Gitschtaler, Andreas Schrenk, Renate Schnitzer, Bernhard Wolkerstorfer and Niklas Baumann (all Boehringer Ingelheim) and Abdel Atrih and Douglas Lamont (both Dundee FingerPrints Proteomics facility) for technical and experimental support, and Mark Pearson, Darryl McConnell and Mark Petronczki (all Boehringer Ingelheim) for critical feedback. We also thank the Diamond Light Source for beamtime (BAG proposal MX14980) and for beamline support at beamline and I24. This work has received funding from Boehringer Ingelheim. Biophysics and drug discovery activities at Dundee were supported by Wellcome Trust strategic awards to Dundee (100476/Z/12/Z and 094090/Z/10/Z, respectively). The Austrian Promotion Agency has funded the project from 2018 until 2021 with the frontrunner grant number 871904 within the general programme.

## Author contributions
C.K. and N.T. designed and synthesised PROTACs, designed and supervised experiments, compiled and analysed data, prepared figures and co-wrote the paper. B.M. designed and supervised experiments, compiled and analysed data, prepared figures and co-wrote the paper. S.W. co-conceived the study and designed, supervised and analysed experiments. M.W. conceived and supervised in vivo work, analysed xenograft experiments and contributed text to the paper. N.M. and J.R. designed and supervised in vivo PK experiments and in vitro ADME assays. M.R. designed and carried out protein crystallography experiments, interpreted, compiled and deposited structural data, prepared figures and contributed text to the manuscript. G.B. designed and supervised protein crystallography experiments, interpreted, compiled and deposited structural data, prepared figures and contributed text to the paper. P.G. and G.Ga. characterised, optimised and scaled up synthesis of compounds and contributed text to the paper. E.D. designed and synthesised PROTACs. R.M. optimised methods for the synthesis of PROTACs. C.W. and V.V. designed and performed cellular experiments. V.V. performed unbiased whole cell proteomics assays. K.R. designed and supervised SPR experiments. M.S. developed SPR assays and acquired and evaluated data. J.E.F. performed enhanced sampling simulations. T.G. designed and supervised cooperativity and ternary complex affinity TR-FRET experiments. Y.C. designed, supervised, and analysed permeability measurements. G.Gr. designed and coordinated biomarker analyses and contributed text to the manuscript. P.C. coordinated and analysed immunohistochemistry experiments. S.H. set up and analysed ex vivo human whole blood assays. N.B. optimised and performed immunohistochemistry experiments. G.Gm. and M.M. designed, performed and analysed NOE experiments, compiled and documented analytical data and contributed text to the paper. M.K. co-conceived the study, co-wrote the manuscript and designed, supervised and analysed experiments. A.C. co-conceived the study, designed and supervised experiments and edited the paper. H.W. co-conceived the study, designed PROTACs, prepared figures and co-wrote the paper. W.F. co-conceived the study, designed PROTACs, designed, supervised and analysed experiments, prepared figures and co-wrote the paper.

## Competing interests
The authors declare the following competing financial interests: The Ciulli laboratory receives or has received sponsored research support from Almirall, Amgen, Amphista therapeutics, Boehringer Ingelheim, Eisai, Merck KGaA, Nurix therapeutics, Ono Pharmaceuticals and Tocris-Bio-Techne. A.C. is a scientific founder, shareholder and consultant of Amphista therapeutics, a company that is developing targeted protein degradation therapeutic platforms. C.K., B.M., S.W., M.W., N.M., G.B., P.G., G.Ga., K.R., M.S., J.E.F., T.G., Y.C., G.Gr., P.C., S.H., N.B., J.R., G.Gm., M.M., M.K. and H.W. are current or former employees of Boehringer Ingelheim. S.W. is now an employee of Merck; E.D. is now an employee of Astra Zeneca. The remaining authors declare no competing interests.

## Additional information

Christiane Kofink [1,5], Nicole Trainor [2,4,5], Barbara Mair [1,5], Simon Wöhrle [1], Melanie Wurm [1], Nikolai Mischerikow[1], Michael J. Roy [2,4], Gerd Bader [1], Peter Greb[1], Géraldine Garavel[1], Emelyne Diers[2], Ross McLennan [2], Claire Whitworth[2], Vesna Vetma [2], Klaus Rumpel [1], Maximilian Scharnweber[1], Julian E. Fuchs [1], Thomas Gerstberger [1], Yunhai Cui [3], Gabriela Gremel[1], Paolo Chetta[1], Stefan Hopf [1], Nicole Budano [1], Joerg Rinnenthal[1], Gerhard Gmaschitz[1], Moriz Mayer[1], Manfred Koegl[1], Alessio Ciulli [2], Harald Weinstabl [1] ✉ & William Farnaby [2] ✉

[1]Boehringer Ingelheim RCV GmbH & Co KG, Vienna, Austria. [2]Centre for Targeted Protein Degradation, School of Life Sciences, University of Dundee, Dundee, UK. [3]Boehringer Ingelheim Pharma GmbH & Co KG, Biberach, Germany. [4]Present address: ACRF Chemical Biology Division, Walter and Eliza Hall Institute, Parkville, VIC, Australia. [5]These authors contributed equally: Christiane Kofink, Nicole Trainor, Barbara Mair. ✉e-mail: harald.weinstabl@boehringer-ingelheim.com; w.farnaby@dundee.ac.uk

