## [Peer Review File · Nature Communications]

A selective and orally bioavailable VHL-recruiting PROTAC achieves SMARCA2 degradation in vivoReviewer #1 (Remarks to the Author):

In the current study, Farnaby et al describe the discovery of an orally bioavailable VHL-recruiting PROTAC which displays a level of selectivity for SMARCA2 degradation over the paralogue SMARCA4. Overall, I found this to be a well written and comprehensive study. Several aspects make the study noteworthy. Firstly, selectivity for SMARCA2 over SMARCA4 was achieved through the intricacies of the respective ternary complexes since the SMARCA binder is equipotent for SMARCA2/4. Secondly, they were able to design in oral bioavailability within the series of compounds. This has been very challenging with VHL based PROTACS although I do have a comment below regarding their claims that this is the first report of an orally bioavailable VHL-recruiting PROTAC. Finally in addition to presenting their data, the authors also share their underlying design principles (e.g. in achieving oral bioavailability) which will be of high interest to those in the field and more broadly to those working in the 'beyond rule of 5' space.

I have the following comments/questions/suggestions:

1. Initially the authors highlight that only CRBN-recruiting PROTACs have achieved oral bio-availability in the clinic which is correct. However, they subsequently go on to assert that 'here we show for the first time that orally bio-available PROTACs can be developed that utilise other E3 ligases, in this case the von Hippel-Lindau (VCB) complex.....'. However, an instance of achieving oral bio-availability with VHL-recruiting PROTACs was recently reported by Xiang et al; Discovery of an orally active VHL-recruiting PROTAC that achieves robust HMGCR degradation and potent hypolipidemic activity in vivo. *Acta Pharm Sin B*. 2021 May;11(5):1300-1314. doi: 10.1016/j.apsb.2020.11.001. While the study Xiang may have certain deficiencies it nevertheless needs to be acknowledged and referenced and the claims in the manuscript adjusted accordingly.

2. The level of selectivity for SMARCA2 over SMARCA4 for compounds 7 and 8 is remarkable and so it is a shame that the selectivity for ACB12 drops to 30-fold. Are the authors able to rationalise why the simple homologation from a 4 to 5 atom linker has such an impact on increasing SMARCA4 degradation potency?

3. The 30-fold SMARCA2 degradation selectivity window achieved with ACB12 may be sufficient for meaningful in vitro studies, but it is potentially limiting in the interpretation of in vivo studies where C_{max} concentrations can readily erode such a margin. In the in vivo studies conducted with ACB12 did the authors probe for SMARCA4 levels (perhaps in blood samples?) to demonstrate that selectivity could be achieved in vivo?

4. Based on the data, I presume that the legend in figure 4a is the wrong way round. i.e. the iv leg is in purple.

5. Page 2, line 16 – use of the word up-cycling is colloquial. Perhaps alternative phrase could be considered.

6. Description of compound synthesis and characterisation in the supporting information was generally of a very high standard with significant provision of ¹H and ¹³C nmr spectra. My only comment is there is no HRMS data on final compounds which would normally be expected as part of their characterisation.

Finally, thanks for the opportunity to review this high quality manuscript. With best wishes,

John Harling

Reviewer #2 (Remarks to the Author):

In this manuscript, Kofink, Trainor, Mair et al. develop and characterize a series of VHL-based PROTACs against SMARCA2, with different degrees of selectivity and potency. Particularly interesting is ACBI2, a novel SMARCA2 (and PBRM1) PROTAC that shows efficacy in vivo using oral administration. According to the authors, ACBI2 is the first orally bioavailable VHL-based PROTAC.

The rational chemical optimization, and overall the structure- and property-based drug design described in the paper are excellent. The number of co-crystals solved is impressive. The work shown is solid, and most of the experiments are of high quality. However, some concerns need to be addressed before this is suitable for acceptance. Especially, there are some claims in which the findings are perhaps oversold, which is not really needed since the quality of the work should stand without the need of being overpraised.

1 - The authors stress many times that they have developed the first orally bioavailable VHL-dependent PROTAC. However, an orally active VHL-recruiting PROTAC against HMGCR was published last year by the Xiang and Zhu labs (doi.org/10.1016/j.apsb.2020.11.001). Is there any reason why this paper was neither cited nor discussed? Aren't there any other examples in the literature?

2 - On page 2 lines 1-2, it would be important to acknowledge (and reference) that there is already a VHL-recruiting PROTAC in clinical trials (DT2216). Although the administration of DT2216 is i.v., it is relevant to contextualize properly that not only CRBN PROTACs are being tested in patients at the moment.

3 - ACBI2 is a degrader of SMARCA2 and PBRM1. This is not mentioned in the title or abstract. In addition, when assessing the phenotypic effects of SMARCA2 degradation, this potential co-founding factor (PBRM1 degradation) is not sufficiently discussed. Can this contribute to explaining the differential effect of ACBI2 compared to genetic studies in SMARCA4-deficient tumors?

4 - There are many experiments in which only quantification is provided, without showing the actual data. It would be extremely important to show the data and/or representative images in: Fig. 2e, Fig. 3c (see comment below), Fig. 3 e-h, Extended data fig. 2a-c (why isn't SMARCA4 included too?), Extended data fig. 3c, and Extended data Fig. 5c.

5 - In fig. 3c, why did the authors treat for >72h? 3 days is the standard when assessing viability using CTG. Especially in cells that grow adherently, 144-192h doesn't seem appropriate. In addition, it seems that different treatment times (144-192h) were used, how are these values then comparable between cell lines? Only the IC50 values are reported (which should be EC50), but the dose curves should be included in an Extended Figure or as Supplementary material.

6 - Why VHL instead of CRBN to develop SMARCA2 PROTACs? Have the authors tried CRBN?

7 - The authors go around the lack of selective SMARCA2 degradation by saying that PBRM1 degradation was expected based on the POI binder (page 8 lines 30-31). As shown several times, degraders can discriminate highly similar targets that don't have differences in affinity for the target ligand alone. In fact, the authors achieve SMARCA2 over SMARCA4 (paralogs) degradation!

8 - It would be valuable to discuss these interesting findings in more detail in the manuscript: (a) For oral dosing, how important was the formulation? This is only mentioned in the methods (15% HP- β -CD). (b) Do other branched linkers show the same NOE trend as the incorporation of a methyl group on the C5 linker? (c) What causes the decrease in efflux ratio by the introduction of the methyl group?

Minor points

- Showing the proteomics data of compound 6 in Fig. 2 instead of in Extend Fig. 4f would be nice. Was SMARCA4 not detected?
- In the discussion, page 10 lines 40-44, the same sentences are repeated.
- Page 2 line 43: you expect to see only the co-crystal structure solved (based on the main text), but then indeed you have a superposition.
- Page 4 last paragraph: the references to the Extended Data figure 2 are not correct (e.g., Ext.DataFig. 2d should be 2f, and 2d-e are not referenced).

- Ext. Data Fig. 5d: I guess you meant lowest/highest value within COLUMN.

Reviewer #3 (Remarks to the Author):

My review has been attached as a pdf as it contains pasted graphics.

(Reviewer #3 Attachment on the following page)

The manuscript by Kofink, et al. reports the design and optimization of a selective and orally available VHL PROTAC. This medicinal chemistry project is important, interesting and well documented. VHL PROTACs will usually have molecular weights at or above 1000 Da, which makes it most challenging to achieve satisfactory oral absorption for them. The manuscript describes how this was achieved by optimization of each of the three parts of the PROTAC using structure-based design. The design of the ligand for the protein of interest in a manner that minimizes the number of hydrogen bond donors was most likely crucial for the success of the project.

The manuscript reports a large volume of work that has been carefully documented to support the reported results and conclusions. The synthetic procedures are a properly described in the Supporting Information. ¹H and ¹³C spectra document the identity and purity of the tested compounds and also allow judgement of their purity. Crystal structures have excellent resolution (<2.25 Å) with clear electron density that allows the ligands of interest to be modeled into the ternary complexes. I did, however, find one part where the data does not support the conclusions presented in the manuscript (cf. comments 1 and 2 below).

Overall, this is an important and well written manuscript that I recommend to be accepted for publication after consideration of the revisions listed below.

Major revisions

1) On pages 5 and 8 it is claimed that branching of the linker by insertion of a methyl group improves the cell permeability of the PROTACs and it is then suggested on page 8 that this is important for achieving good oral absorption for ACBI2 (22%). However, the Caco-2 data for the matched molecular pairs **7** and **8**, and **10** and ACBI2, each of which only differ by the addition of a branching methyl group in **8** and ACBI2 do not support this conclusion. The permeabilities reported in Extended Data Table 4 for these compounds are

	Solubility pH 4.5 (µg/mL)	Solubility pH 6.8 (µg/mL)	clogP	Caco-2 P _{ab} (10 ⁻⁶ cm/s)	Caco-2 Efflux ratio	Modified Caco-2 P _{ab} (10 ⁻⁶ cm/s)	Modified Caco-2 Efflux ratio	Modified Caco-2 + PGPI (Zosuquidar) P _{ab} (10 ⁻⁶ cm/s)	Modified Caco-2 + PGPi (Zosuquidar) Efflux ratio
compound 7	<1 (n = 1)	<1 (n = 1)	8.9	<0.5 (n = 3)	-	2 ± 1 (n = 3)	79 ± 10 (n = 3)	10 ± 1 (n = 3)	3 ± 1 (n = 3)
compound 8	18 (n = 3)	<1 (n = 3)	9.2	6 ± 2 (n = 3)	5 ± 2 (n = 3)	3 ± 1 (n = 3)	52 ± 11 (n = 3)	12 ± 2 (n = 3)	2 ± 0.2 (n = 3)

compound 10	<1	<1	9.9	1.4 ± 0.5	7 ± 1	7 ± 5	30 ± 20	11 ± 3	2 ± 0.2
----------------	----	----	-----	-----------	-------	-------	---------	--------	---------

	(n = 8)	(n = 8)		(n = 4)	(n = 4)	(n = 4)	(n = 4)	(n = 4)	(n = 4)
ACBI2	<1	<1	10.3	2 ± 0.2	5 ± 1	5 ± 1	9 ± 2	8 ± 4	2 ± 1
	(n = 4)	(n = 4)		(n = 4)	(n = 4)	(n = 3)	(n = 3)	(n = 3)	(n = 3)

As documented in the last two columns the efflux inhibited, passive permeabilities are not statistically significant for the two matched pairs, nor between the pairs. There are some differences in efflux ratios (cf. third column from the right) within the pairs, suggesting introduction of the methyl group to result in a minor reduction of efflux. However, the major reduction in efflux is obtained by replacing the alkoxy-linker found in **7** and **8** by the aliphatic one in **10** and ACBI2. For instance, **8** and ACBI2, both of which have branched linkers have ERs of 52 and 9 in the modified Caco2 assay, respectively. However, ERs do not differ between these compounds in the standard Caco2 assay (column 6 from the left). Thus, it is the replacement of the alkoxy-linker of **8** by the aliphatic one in ACBI2 that is the major reason for the satisfactory oral absorption of ACBI2, not the branching of the linker (at least based on comparison of ERs in the modified Caco2 assay). This makes the discussion of folded conformations as a reason for achieving a good oral absorption for ACBI of limited interest. Consequently, this part of the manuscript should be revised. The computational and NMR studies conducted for **10** and ACBI2 are then less relevant, or maybe irrelevant.

2) The presence of long-range NOEs for ACBI2, which are absent in compound **10**, are used to support that ACBI2 adopts folded conformations. This comparison is based on single measurements with a mixing time of 700 ms that gave weak signals for ACBI2 but no signals for **10**. Measurements appear to have been done close to the NOE zero crossover point. The absence of a NOE may result from this fact, or from differences in T1 and T2 relaxation. Therefore, single measurements do not provide reliable results

If the authors would like to keep the discussion of differences in NOEs between ACBI2 and **10** in the manuscript, spectra should be acquired using several different mixing times for both compounds. This will allow determination of NOE build ups for the selected long range distances for both compounds, which provide reliable proof for structural proximity of protons located far from each other in the PROTACs.

Minor revisions

- 3) In the summary paragraph it is pointed out that design of orally available PROTACs is challenging due to their high molecular weight and often extreme values for other physicochemical properties. Here it is appropriate to cite the two papers that first pointed this out in 2019, i.e. the articles by Edmondson, et al. (<https://doi.org/10.1016/j.bmcl.2019.04.030>) and Maple, et al. (DOI: 10.1039/c9md00272c).
- 4) Inclusion of the structure of compound 4 in Figure 1c would help the reader and is easy to do.
- 5) It is very difficult to gain any information from the ternary complexes shown in Figures 2b and 2d. They should be enlarged by 50% or more.
- 6) The curves for i.v. and p.o. in Figure 3c can not be distinguished from each other, apart from by the obvious fact that the one for p.o. will be below the i.v. one. Use of different colors would convey this information in a better way to the reader.
- 7) PDB IDs for the crystal structures should be included in the legends of the figure where the structures are presented.

Response to reviewers:

Reviewer 1:

In the current study, Farnaby et al describe the discovery of an orally bioavailable VHL-recruiting PROTAC which displays a level of selectivity for SMARCA2 degradation over the paralogue SMARCA4. Overall, I found this to be a well written and comprehensive study. Several aspects make the study noteworthy. Firstly, selectivity for SMARCA2 over SMARCA4 was achieved through the intricacies of the respective ternary complexes since the SMARCA binder is equipotent for SMARCA2/4. Secondly, they were able to design in oral bioavailability within the series of compounds. This has been very challenging with VHL based PROTACS although I do have a comment below regarding their claims that this is the first report of an orally bioavailable VHL-recruiting PROTAC. Finally in addition to presenting their data, the authors also share their underlying design principles (e.g. in achieving oral bioavailability) which will be of high interest to those in the field and more broadly to those working in the 'beyond rule of 5' space.

I have the following comments/questions/suggestions:

We are excited to read this positive response and that the key aspects of our study have been recognised by the reviewer. We have endeavoured to address all of the feedback, point-by-point, below.

1. Initially the authors highlight that only CRBN-recruiting PROTACs have achieved oral bio-availability in the clinic which is correct. However, they subsequently go on to assert that 'here we show for the first time that orally bio-available PROTACs can be developed that utilise other E3 ligases, in this case the von Hippel-Lindau (VCB) complex.....'. However, an instance of achieving oral bio-availability with VHL-recruiting PROTACs was recently reported by Xiang et al; Discovery of an orally active VHL-recruiting PROTAC that achieves robust HMGCR degradation and potent hypolipidemic activity in vivo. Acta Pharm Sin B. 2021 May;11(5):1300-1314. doi: 10.1016/j.apsb.2020.11.001. While the study Xiang may have certain deficiencies it nevertheless needs to be acknowledged and referenced and the claims in the manuscript adjusted accordingly.

We appreciate the reviewer's comments on this and indeed Xiang et al previously reported VHL PROTACs with oral exposure. However, whilst this was a breakthrough for the field, quantifiable bioavailability was not demonstrated in that study and/because no i.v. dosing data was produced. Furthermore, broader learnings for how VHL based PROTACs and large bifunctional may be optimised to achieve oral bioavailability and efficacy and how this can be quantified have not previously been presented. To address the reviewers comments we have referenced the Xiang paper and qualified our claims as follows on page 2, lines 4-6 (the word 'first' has been removed where used in this context):

Recently, a study by Xiang et al demonstrated oral exposure for VHL PROTACs, but did not quantify oral bioavailability⁸. We present here a breakthrough in optimising VHL-based PROTACs to obtain quantifiable oral bioavailability in addition to in vivo efficacy.

2. The level of selectivity for SMARCA2 over SMARCA4 for compounds 7 and 8 is remarkable and so it is a shame that the selectivity for ACB2 drops to 30-fold. Are the authors able to rationalise why the simple homologation from a 4 to 5 atom linker has such an impact on increasing SMARCA4 degradation potency?

We thank the reviewer for their insightful comment and seek to clarify as detailed below. We indeed observe higher selectivity in cells for SMARCA2 vs SMARCA4 degradation for compounds 8 and 9 (previously 7 and 8) compared with ACBI2. Structurally, compounds 8 and 9 differ from ACBI2 in two aspects, in that they contain an ether linkage, whereas ACBI2 has an all-carbon linker and in addition, ACBI2 has a longer linker by one atom. In an attempt to understand differences in selectivity, we were able to generate high resolution ternary co-crystals structures that enable hypotheses to explain the selectivity observed for compounds in the ether series. Further, TR-FRET data (extended data table 3) supports that there is a difference in both ternary affinity and cooperativity for SMARCA2 over SMARCA4 for these compounds. ACBI2, however, demonstrates a reduced window of preference in the same TR-FRET assay for forming ternary complexes with SMARCA2 bromodomain vs SMARCA4 bromodomain. Given this and the difference in linker composition we would hypothesise the likelihood of a different ternary complex arrangement being formed for ACBI2 is high^{17,18}. Unfortunately, we were not able to solve ternary co-crystal structures to ascertain whether ACBI2 to support this further.

To address this point specifically in the main text we have added two sections of text that address both the broader and more specific aspects of this as follows:

Page 6, lines 18-21:

Findings in multiple protein degradation projects and recent publications have demonstrated that changing the linker even by only one atom can have a significant influence on molecular properties e.g. due to a different three-dimensional conformation of the molecule¹⁸, but also selectivity by inducing different ternary complexes^{19,20}.

Page 12 lines 1-11:

We observed higher selectivity for compounds from the ether series (e.g. compound **10**) compared with ACBI2. Structurally, these compounds differ both in composition (e.g. ether vs all carbon) and length. Both biochemical and ternary co-crystal structures illuminate possible contributory factors for enhanced selectivity of the ether series. For example, ether series compounds demonstrate greater differences in cooperativity between SMARCA2 and SMARCA4 compared with ACBI2 (Extended Data Table 3), and compound **10** features a key ternary complex binding interaction with SMARCA2 specific residue Gln1469 (Figure **3b**, PDB: 7Z76). Given this and the difference in linker composition we would hypothesise the likelihood of a different ternary complex arrangement being formed for ACBI2 is high, contributing to its lower SMARCA2 selectivity compared with ether series molecules, though this was accepted as a trade-off due to the improved oral bioavailability of ACBI2. Unfortunately, we were not able to solve a ternary co-crystal structures to ascertain ACBI2's ternary binding mode.

3. The 30-fold SMARCA2 degradation selectivity window achieved with ACBI2 may be sufficient for meaningful in vitro studies, but it is potentially limiting in the interpretation of in vivo studies where C_{max} concentrations can readily erode such a margin. In the in vivo studies conducted with ACBI2 did the authors probe for SMARCA4 levels (perhaps in blood samples?) to demonstrate that selectivity could be achieved in vivo?

We agree that in vivo selectivity is important. We had previously observed that compound 6 did not show degradation of either SMARCA2 or SMARCA4 in lung tissue taken from NCI-H1568 xenograft-bearing mice, indicating that the compound might not be fully cross-reactive in mouse. We therefore addressed this by carrying out ex vivo degradation assays in a human whole blood assay, arguably more translationally relevant in any case. This data is shown in in Fig. 4i and supports that ACBI2 has

significant window of selectivity for SMARCA2 vs SMARCA4, where we do not see more than 50% SMARCA4 degradation up to a top concentration of 1 μ M.

4. Based on the data, I presume that the legend in figure 4a is the wrong way round. i.e. the iv leg is in purple.

We thank the reviewer for spotting this error and have corrected it in the revised version of the figure.

5. Page 2, line 26 – use of the word up-cycling is colloquial. Perhaps alternative phrase could be considered.

We thank the reviewer for the suggestion and have changed the wording to “PROTACs are often made by the **conversion of** existing protein of interest (POI) binder”

(N.B. now on page 2, line 24)

6. Description of compound synthesis and characterisation in the supporting information was generally of a very high standard with significant provision of ¹H and ¹³C nmr spectra. My only comment is there is no HRMS data on final compounds which would normally be expected as part of their characterisation.

We have now included HRMS data for final compounds as requested in the supporting information (page 80 onwards).

Finally, thanks for the opportunity to review this high quality manuscript. With best wishes,

John Harling

Reviewer 2:

In this manuscript, Kofink, Trainor, Mair et al. develop and characterize a series of VHL-based PROTACs against SMARCA2, with different degrees of selectivity and potency. Particularly interesting is ACBI2, a novel SMARCA2 (and PBRM1) PROTAC that shows efficacy in vivo using oral administration. According to the authors, ACBI2 is the first orally bioavailable VHL-based PROTAC. The rational chemical optimization, and overall the structure- and property-based drug design described in the paper are excellent. The number of co-crystals solved is impressive. The work shown is solid, and most of the experiments are of high quality. However, some concerns need to be addressed before this is suitable for acceptance. Especially, there are some claims in which the findings are perhaps oversold, which is not really needed since the quality of the work should stand without the need of being overpraised.

We thank you reviewer for their feedback. We address their comments point-by-point below and have paid particular attention to tone down claims and language where appropriate, as advised.

1 - The authors stress many times that they have developed the first orally bioavailable VHL-dependent PROTAC. However, an orally active VHL-recruiting PROTAC against HMGCR was published last year by the Xiang and Zhu labs (doi.org/10.1016/j.apsb.2020.11.001). Is there any reason why this paper was neither cited nor discussed? Aren't there any other examples in the literature?

We acknowledge the reviewer's feedback and have endeavoured to address specifically the significance of the Xiang paper. Indeed Xiang et al previously reported VHL PROTACs with oral exposure. However, whilst this was a breakthrough for the field, quantifiable bioavailability was not demonstrated in that study and/because no i.v. dosing data was produced. Furthermore, broader learnings for how VHL based PROTACs and large bifunctional may be optimised to achieve oral bioavailability and efficacy and how this can be quantified have not previously been presented. . To address the reviewers comments we have referenced the Xiang paper and qualified our claims as follows on page 2, line 4-6 (the word 'first' has been removed where used in this context'):

Recently, a study by Xiang et al demonstrated oral exposure for VHL PROTACs, but did not quantify oral bioavailability⁸. We present here a breakthrough in optimising VHL-based PROTACs to obtain quantifiable oral bioavailability in addition to in vivo efficacy.

2 - On page 2 lines 1-2, it would be important to acknowledge (and reference) that there is already a VHL-recruiting PROTAC in clinical trials (DT2216). Although the administration of DT2216 is i.v., it is relevant to contextualize properly that not only CRBN PROTACs are being tested in patients at the moment.

We have now included a reference to this compound and on Page 2, line 2-3 text was changed as follows:

To best of our knowledge, only one VHL-recruiting PROTAC (DT2216) administered via intravenous infusion is currently in clinical trials^{5, 6}.

3 - ACBI2 is a degrader of SMARCA2 and PBRM1. This is not mentioned in the title or abstract. In addition, when assessing the phenotypic effects of SMARCA2 degradation, this potential co-founding factor (PBRM1 degradation) is not sufficiently discussed. Can this contribute to explaining the differential effect of ACBI2 compared to genetic studies in SMARCA4-deficient tumors?

To address this point we have now included the following sentence in the summary paragraph, page 2 lines 12-16:

ACBI2 is a full degrader of SMARCA2 and PBRM1. Contributions of PBRM1 degradation to phenotypes can therefore not be excluded. However, it has previously been shown with ACBI1, a degrader of SMARCA2, SMARCA4 and PBRM1, that the anti-proliferative effects in NCI-H1568 (SMARCA4 deficient) cells can be negated by overexpression of SMARCA2 but not PBRM1⁹.

4 – There are many experiments in which only quantification is provided, without showing the actual data. It would be extremely important to show the data and/or representative images in: Fig. 2e, Fig. 3c (see comment below), Fig.3 e-h, Extended data fig. 2a-c (why isn't SMARCA4 included too?), Extended data fig. 3c, and Extended data Fig. 5c.

We thank the reviewer for the suggestion – we have now included a single source data file consistent with journal guidance that addresses this request. Additionally, representative IHC images for all biomarker measurements and for the mentioned WB/WES experiments are included in the respective figures, with uncropped blots in the source data file. In Extended Fig. 2a-c, we used SMARCA4-deficient NCI-H1568 cells in preparation for subsequent in vivo efficacy experiments, therefore, SMARCA4 is not detectable in these experiments and is hence not included. We have also added the individual curves for Fig. 4c (which we believe the reviewer meant instead of 3c) and Extended Fig. 5d.

5 - In fig. 3c, why did the authors treat for >72h? 3 days is the standard when assessing viability using CTG. Especially in cells that grow adherently, 144-192h doesn't seem appropriate. In addition, it seems that different treatment times (144-192h) were used, how are these values then comparable between cell lines? Only the IC₅₀ values are reported (which should be EC₅₀), but the dose curves should be included in an Extended Figure or as Supplementary material.

The CTG assays in Fig 4c (which we believe the reviewer refers to) were individually optimized for each cell line to reach a common confluency endpoint to allow for comparison, i.e. taking their doubling time into account. Moreover, we have noticed that 72h assays can underestimate the full effect size, in particular with compounds acting on epigenetic targets. Hence, we typically run CTG assays for >72h in our organization. We have added the dose response curves for Fig. 4c and Extended Fig. 5d and have also changed IC₅₀ to EC₅₀ throughout the manuscript.

6 – Why VHL instead of CRBN to develop SMARCA2 PROTACs? Have the authors tried CRBN?

We fully appreciate that it is logical to survey more than one ligase when pursuing a target for degradation and one of our motivations for publishing this study is indeed to widen the available knowledge base for a broader selection of ligases. To date, there has been a large degree of focus on the in vivo translation of CRBN based degraders, so we expect this study provides useful learnings for the field when working outside of a CRBN focus. In this case our early degraders were VHL based and the scope of this study is to detail learnings in how VHL based degraders can be progressed. Nevertheless, we will endeavor to share our learnings in the future around degraders based on CRBN (or other ligases).

7 - The authors go around the lack of selective SMARCA2 degradation by saying that PBRM1 degradation was expected based on the POI binder (page 8 lines 30-31). As shown several times, degraders can discriminate highly similar targets that don't have differences in affinity for the target ligand alone. In fact, the authors achieve SMARCA2 over SMARCA4 (paralogs) degradation!

We agree that selectivity can be achieved beyond that of the constituent ligands and as the reviewer points out and see this study as another strong example of this with respect to achieving SMARCA2 selectivity over SMARCA4. Nevertheless, there is no guarantee that such selectivity can or always will be achieved when using a non-selective ligand, as has also been shown including with our previous publication detailing PROTACs that degrade SMARCA2/4 and PBRM1 (Farnaby, Koegl, et al. *Nat Chem Biol.* 15, 672-680 (2019)). It is therefore logical to anticipate that degradation of PBRM1 is a reasonably likely outcome in this case, just as for some compounds in this study we observe no selectivity for SMARCA2 over SMARCA4 either.

8 – It would be valuable to discuss these interesting findings in more detail in the manuscript:

(a) For oral dosing, how important was the formulation? This is only mentioned in the methods (15% HP-β-CD).

We did not conduct systematic formulation studies in mouse. We use HP-β-CD in the concentration range of 10-20% as standard approach to formulate poorly soluble compounds for in vivo studies. Of note, this agent was also used by Xiao *et al.* (Xiao et al, *Nature*, 601, 434-439 (2021)) to formulate a VHL-based SMARCA PROTAC for intravenous administration.

(b) Do other branched linkers show the same NOE trend as the incorporation of a methyl group on the C5 linker?

NOESY experiments of compound 30 in CDCl₃, which has the dimethyl branched C5 linker also shows long range NOEs indicated for ACBI2, see also Figure S5 in the supplementary Information.

(c) What causes the decrease in efflux ratio by the introduction of the methyl group?

Following suggestion of the reviewers we have retested in triplicate key molecules in this manuscript in the modified caco assay. We find that there is a significant difference between the ether and all carbon series and have commented on this being in line with classical TPSA reduction. Larger variability within this assay prevents firm conclusions on a link between efflux and linker branching directly between compound 11 and ACBI2, however the differences in in solution conformation and oral bioavailability are robustly demonstrated. We have adjusted the text in this section accordingly (yellow highlight) on page 8, lines 7 to 12.

Minor points

- Showing the proteomics data of compound 6 in Fig. 2 instead of in Extend Fig. 4f would be nice. Was SMARCA4 not detected?

The cell line used here (NCI-H1568) is SMARCA4 deficient. We have now included this experiment in main Figure 2d. Of note, compound numbering had to be changed accordingly: former compound 11 is now 7, 10 is 11, 9 is 10, 8 is 9 and 7 is 8.

- In the discussion, page 10 lines 40-44, the same sentences are repeated.

We thank the reviewer for spotting this error and have corrected it in the revised version of the manuscript.

- Page 2 line 43: you expect to see only the co-crystal structure solved (based on the main text), but then indeed you have a superposition.

We agree with the reviewer and have added compound 4 and its interactions as standalone figure panel in 1d.

- Page 4 last paragraph: the references to the Extended Data figure 2 are not correct (e.g., Ext.Data Fig. 2d should be 2f, and 2d-e are not referenced).

Again, we thank the reviewer for spotting the error and have corrected it in the revised version of the manuscript.

- Ext. Data Fig. 5d: I guess you meant lowest/highest value within COLUMN.

Once more, we thank the reviewer for spotting the error, we have corrected it now.

Reviewer 3:

The manuscript by Kofink, et al. reports the design and optimization of a selective and orally available VHL PROTAC. This medicinal chemistry project is important, interesting and well documented. VHL PROTACs will usually have molecular weights at or above 1000 Da, which makes it most challenging to achieve satisfactory oral absorption for them. The manuscript describes how this was achieved by optimization of each of the three parts of the PROTAC using structure-based design. The design of the ligand for the protein of interest in a manner that minimizes the number of hydrogen bond donors was most likely crucial for the success of the project.

The manuscript reports a large volume of work that has been carefully documented to support the reported results and conclusions. The synthetic procedures are a properly described in the

Supporting Information. ¹H and ¹³C spectra document the identity and purity of the tested compounds and also allow judgement of their purity. Crystal structures have excellent resolution (<2.25 Å) with clear electron density that allows the ligands of interest to be modeled into the ternary complexes. I did, however, find one part where the data does not support the conclusions presented in the manuscript (cf. comments 1 and 2 below). Overall, this is an important and well written manuscript that I recommend to be accepted for publication after consideration of the revisions listed below.

We are delighted to receive this positive response from the reviewer. All of their points have been addressed below, point-by-point. In particular we have repeated efflux measurements and amended our claims to address point 1. We have also provided additional NMR data to support our conclusions in point 2. We hope this serves to address the major points of recommendation.

Major revisions

1) On pages 5 and 8 it is claimed that branching of the linker by insertion of a methyl group improves the cell permeability of the PROTACs and it is then suggested on page 8 that this is important for achieving good oral absorption for ACBI2 (22%). However, the Caco-2 data for the matched molecular pairs **7** and **8**, and **10** and ACBI2, each of which only differ by the addition of a branching methyl group in **8** and ACBI2 do not support this conclusion. The permeabilities reported in Extended Data Table 4 for these compounds are

	Solubility pH 4.5 (µg/mL)	Solubility pH 6.8 (µg/mL)	clogP	Caco-2 P _{app} (10 ⁻⁶ cm/s)	Caco-2 Efflux ratio	Modified Caco-2 P _{app} (10 ⁻⁶ cm/s)	Modified Caco-2 Efflux ratio	Modified Caco-2 + PGPI (Zosuquidar) P _{app} (10 ⁻⁶ cm/s)	Modified Caco-2 + PGPI (Zosuquidar) Efflux ratio
--	---------------------------------	---------------------------------	-------	---	------------------------	---	------------------------------------	---	--

compound 7	<1 (n = 1)	<1 (n = 1)	8.9	<0.5 (n = 3)	-	2 ± 1 (n = 3)	79 ± 10 (n = 3)	10 ± 1 (n = 3)	3 ± 1 (n = 3)
compound 8	18 (n = 3)	<1 (n = 3)	9.2	6 ± 2 (n = 3)	5 ± 2 (n = 3)	3 ± 1 (n = 3)	52 ± 11 (n = 3)	12 ± 2 (n = 3)	2 ± 0.2 (n = 3)

compound 10	<1	<1	9.9	1.4 ± 0.5	7 ± 1	7 ± 5	30 ± 20	11 ± 3	2 ± 0.2
----------------	----	----	-----	-----------	-------	-------	---------	--------	---------

	(n = 8)	(n = 8)		(n = 4)	(n = 4)	(n = 4)	(n = 4)	(n = 4)	(n = 4)
ACBI2	<1 (n = 4)	<1 (n = 4)	10.3	2 ± 0.2 (n = 4)	5 ± 1 (n = 4)	5 ± 1 (n = 3)	9 ± 2 (n = 3)	8 ± 4 (n = 3)	2 ± 1 (n = 3)

As documented in the last two columns the efflux inhibited, passive permeabilities are not statistically significant for the two matched pairs, nor between the pairs. There are some differences in efflux ratios (cf. third column from the right) within the pairs, suggesting introduction of the methyl group to result in a minor reduction of efflux. However, the major reduction in efflux is obtained by replacing the alkoxy-linker found in **7** and **8** by the aliphatic one in **10** and ACBI2. For instance, **8** and ACBI2, both of which have branched linkers have ERs of 52 and 9 in the modified Caco2 assay, respectively. However, ERs do not differ between these compounds in the standard Caco2 assay (column 6 from the left). Thus, it is the replacement of the alkoxy-linker of **8** by the aliphatic one in ACBI2 that is the major reason for the satisfactory oral absorption of ACBI2, not the branching of the linker (at least based on comparison of ERs in the modified Caco2 assay). This makes the discussion of folded conformations as a reason for achieving a good oral absorption for ACBI of limited interest. Consequently, this part of the manuscript should be revised. The computational and NMR studies conducted for **10** and ACBI2 are then less relevant, or maybe irrelevant.

We highly appreciate this important feedback from the reviewer and the opportunity to modify. We have now repeated testing (in triplicate) of compounds in the modified CaCo assay and included updated means and standard deviations in the data table. The reviewer is correct that the large errors from this assay prevent firm conclusions when comparing efflux ratios for compound **11** (formerly **10**) and ACBI2. However, we also agree with the reviewer that there is a significant difference in efflux when moving from ether to all carbon based series, which can be explained classically via TPSA reduction.

Despite the lack of clarity in the CaCo data, we do observe robust improvement in in vivo bioavailability between compound **11** and ACBI2 and thus believe any information pertaining to differences in physicochemical properties or in solution behaviour are highly relevant data for wider learning in the field. To this end we have extended our NMR studies (see point below).

To address this point by the review we have amended the text in this section (yellow highlighted in the manuscript) to tone down claims and represent the data included. This can be seen on page 8 lines 7-12.

2) The presence of long-range NOEs for ACBI2, which are absent in compound **10**, are used to support that ACBI2 adopts folded conformations. This comparison is based on single measurements with a mixing time of 700 ms that gave weak signals for ACBI2 but no signals for **10**. Measurements appear to have been done close to the NOE zero crossover point. The absence of a NOE may result from this fact, or from differences in T1 and T2 relaxation. Therefore, single measurements do not provide reliable results. If the authors would like to keep the discussion of differences in NOEs between ACBI2 and **10** in the manuscript, spectra should be acquired using several different mixing times for both compounds. This will allow determination of NOE build ups for the selected long range distances for both compounds, which provide reliable proof for structural proximity of protons located far from each other in the PROTACs.

We thank the reviewer for these comments and appreciate the opportunity to expand this data set to support our conclusions. We followed the reviewers' comments and added different NOE mixing times from 0.1 to 1.0 seconds and show the build-up curves of the three NOEs indicated by arrows in Figure 4b in the Supplements in Figure S3. We also included T1 relaxation times in the Supplementary Section of selected well resolved signals of compound **11** (formerly **10**) and ACBI2 which are very similar (NMR Table2). Further since compound **11** is close to the NOE zero crossing in CDCl₃, whereas ACBI2 is still mostly in the positive NOE regime at 600 MHz we have included 1D selective ROESYs with the tBu group selectively inverted which have no zero crossing. Compound **11**

has a lower solubility in CDCl_3 than ACBI2 which we compensated through much longer acquisition times in order to achieve the same S/N ratios of a chosen ROE enhancement (proton 37). We have added the 1D ROE traces to the supplementary information in Figure S4 and rephrased the supplement accordingly.

Minor revisions

3) In the summary paragraph it is pointed out that design of orally available PROTACs is challenging due to their high molecular weight and often extreme values for other physicochemical properties. Here it is appropriate to cite the two papers that first pointed this out in 2019, i.e. the articles by Edmondson, et al. (<https://doi.org/10.1016/j.bmcl.2019.04.030>) and Maple, et al. (DOI:10.1039/c9md00272c).

We have now added these references as requested – references 3 and 4, page 1, line 43.

4) Inclusion of the structure of compound 4 in Figure 1c would help the reader and is easy to do.

We have now added compound 4 and its interactions as standalone figure panel in 1d.

5) It is very difficult to gain any information from the ternary complexes shown in Figures 2b and 2d. They should be enlarged by 50% or more.

We thank the reviewer for pointing this out and have enlarged the figure panels.

6) The curves for i.v. and p.o. in Figure 3c cannot be distinguished from each other, apart from by the obvious fact that the one for p.o. will be below the i.v. one. Use of different colors would convey this information in a better way to the reader.

We thank the reviewer for the suggestion and have now improved the contrast between the colours.

7) PDB IDs for the crystal structures should be included in the legends of the figure where the structures are presented.

We have included the PDB IDs in main text, figure and table legends now.

Reviewer #1 (Remarks to the Author):

Farnaby and colleagues have resubmitted their SMARCA PROTAC manuscript following peer review. Their responses to my review comments have been both positive and comprehensive, and I am happy with the modifications that they have made to the manuscript. Responses to comments from other reviewers appear to be similarly comprehensive. I have no further comments and recommend publication.

Reviewer #2 (Remarks to the Author):

The most important concerns pointed out by the reviewers have been addressed. Happy to recommend the publication of the manuscript in Nature Communications, congratulations to the authors.

Reviewer #3 (Remarks to the Author):

As I pointed out in review of the first version of this manuscript it reports important, interesting and well documented results, i.e. the design and optimization of a selective and orally available VHL PROTAC. The authors have handled my original suggestions for revision well and I therefore recommend that it is accepted after handling of the following revisions:

1) I really struggled to understand the paragraph starting at the bottom of page 2. The reason for this is that I got stuck on the text on lines 40-42: we first characterised the molecular interactions that are mandatory for BD binding using a high-resolution crystal structure of a close analogue of "compound 26"¹⁶ (Figure 1a, PDB: 5FH7 shows related "compound 18" co-crystallized with PBRM1BD5). It would have helped tremendously if structure of "compound 18" had been shown together with "compound 26" in Figure 1a, and if the text in the parenthesis is adjusted. As this text is written now one expects to find the co-crystal structure of "compound 18" in Figure 1a.

2) Page 4, line 31. Compound 6 should have 6 in boldface.

3) Page 8, lines 2-7. I believe that this part may overemphasize the effect of PROTAC folding on cell permeability and oral absorption, and omits to point out that CRBN PROTACs are in general closer to oral druggable space as compared to VHL PROTACs (cf. refs 3 and 4). In addition, the phrasing there is a perception that it is more difficult to discover oral VHL than CRBN PROTACs is somewhat misleading. Pike, et al (ref 24) provide experimental data that reveals this to be true for a large number of CRBN and VHL PROTACs. Some minor adjustments of of this paragraph would be appropriate.

4) On page 8 the authors have revised the text to fit the Caco-2 permeability data for compounds 6-11 and ACBI2. Extended Data Table 4 shows that ACBI2 has lower efflux than 11, while their efflux inhibited permeabilities do not differ significantly. When reading the text on page 8 one may still get the impression that the improved oral availability of ACBI2 over 11, which correlates to an increased folding as determined by NMR and MD simulations, originates from a higher passive cell permeability of ACBI2. It is better to state clearly that in this case the folding determined by NMR and MD simulations appears to influence the efflux ratio, not the passive cell permeability. In contrast, a manuscript recently posted on ChemRxiv

(<https://chemrxiv.org/engage/chemrxiv/article-details/6271528b5b9009189a25f318>) concluded that the differences in passive permeability of three CRBN PROTACs correlated to their folding.

5) Page 11, line 46. "These degree" should read "The degree".

6) The headings of the Extended Data Tables are placed after the tables. It would be better if they are placed at the top of the tables. In addition, it is difficult for the non-specialist to understand what data is given in the columns of the tables. For instance, I struggled with Extended Data Table 1; e.g what is the VCB+SMARCA2 KD in column 4 as compared to the SMARCA2 KD given in column 10. Considering that such a large volume of important data is presented in the Extended Data I strongly recommend that the authors spend time inserting footnotes that clearly describe what data is found in the tables.

Response to reviewers:

Reviewer #1 (Remarks to the Author):

Farnaby and colleagues have resubmitted their SMARCA PROTAC manuscript following peer review. Their responses to my review comments have been both positive and comprehensive, and I am happy with the modifications that they have made to the manuscript. Responses to comments from other reviewers appear to be similarly comprehensive. I have no further comments and recommend publication.

Many thanks to Reviewer 1 for taking the time to read our manuscript and for their feedback throughout the peer review process.

Reviewer #2 (Remarks to the Author):

The most important concerns pointed out by the reviewers have been addressed. Happy to recommend the publication of the manuscript in Nature Communications, congratulations to the authors.

Many thanks to Reviewer 2 for taking the time to read our manuscript and for their feedback throughout the peer review process.

Reviewer #3 (Remarks to the Author):

As I pointed out in review of the first version of this manuscript it reports important, interesting and well documented results, i.e. the design and optimization of a selective and orally available VHL PROTAC. The authors have handled my original suggestions for revision well and I therefore recommend that it is accepted after handling of the following revisions:

1) I really struggled to understand the paragraph starting at the bottom of page 2. The reason for this is that I got stuck on the text on lines 40-42: we first characterised the molecular interactions that are mandatory for BD binding using a high-resolution crystal structure of a close analogue of “compound 26”¹⁶ (Figure 1a, PDB: 5FH7 shows related “compound 18” co-crystalized with PBRM1BD5). It would have helped tremendously if structure of “compound 18” had been shown together with “compound 26” in Figure 1a, and if the text in the parenthesis is adjusted. As this text is written now one expects to find the co-crystal structure of “compound 18” in Figure 1a.

We thank the reviewer for bringing this to our attention and agree that more clarity is needed to improve the readability of this paragraph and to distinguish which of the two literature compounds is associated with the PDB entry 5FH7. We have changed the wording (now p3 lines 4-7) as follows and we have included “compound 18” in Figure 1a.

...we first characterised the molecular interactions that are mandatory for BD binding using a high-resolution crystal structure of “compound 18” (PDB: 5FH7, chemical structure in Figure 1a). We found the halogen bond to the Met731^{PBRM1} backbone (BB) carbonyl and the hydrogen bonding interaction between Asn739^{PBRM1} and the quinazolinone core to be indispensable and witnessed SAR findings leading to “compound 26”.

2) Page 4, line 31. Compound 6 should have 6 in boldface.

Thank you to the reviewer for spotting this error – we have corrected the formatting.

3) Page 8, lines 2-7. I believe that this part may overemphasize the effect of PROTAC folding on cell permeability and oral absorption, and omits to point out that CRBN PROTACs are in general closer to oral druggable space as compared to VHL PROTACs (cf. refs 3 and 4). In addition, the phrasing there is a perception that it is more difficult to discover oral VHL than CRBN PROTACs is somewhat misleading. Pike, et al (ref 24) provide experimental data that reveals this to be true for a large number of CRBN and VHL PROTACs. Some minor adjustments of of this paragraph would be appropriate.

We thank the reviewer for feedback on this point. We have adapted the phrasing to reflect that CRBN PROTACs tend more towards 2D physicochemical profiles closer to classical druggable space. We also wish to refrain from overstating the effects of PROTAC folding and have altered the text as highlighted below. Pike et al is now cited in the Introduction section (p2 line 5, Ref 5). The changes can now be found on p5, lines 16-17:

Due to the differences of the respective E3 ligase binders, VHL PROTACs tend to have 2D physicochemical properties further from classical oral druggable space as compared with CRBN PROTACs^{3,4}. Nevertheless, it has been shown that in some cases PROTACs can adopt more compact 3D conformations that yield 3D polar surface and radius of gyration more consistent with that required for permeability^{18,19}

4) On page 8 the authors have revised the text to fit the Caco-2 permeability data for compounds 6-11 and ACBI2. Extended Data Table 4 shows that ACBI2 has lower efflux than 11, while their efflux inhibited permeabilities do not differ significantly. When reading the text on page 8 one may still get the impression that the improved oral availability of ACBI2 over 11, which correlates to an increased folding as determined by NMR and MD simulations, originates from a higher passive cell permeability of ACBI2. It is better to state clearly that in this case the folding determined by NMR and MD simulations appears to influence the efflux ratio, not the passive cell permeability. In contrast, a manuscript recently posted on ChemRxiv (<https://chemrxiv.org/engage/chemrxiv/article-details/6271528b5b9009189a25f318>) concluded that the differences in passive permeability of three CRBN PROTACs correlated to their folding.

We agree with the reviewer regarding their interpretation of our Caco-2 data for compound **11** and ACBI2. We have amended the paragraph to include a focus on the greater conformational restraint of ACBI2, which could lead to an improvement in the efflux ratio and how this may influence oral bioavailability. Please see P5, line 32:

Taken with measurements in Caco-2 cells, in which **11** and ACBI2 show similar passive permeabilities in the presence of an efflux inhibitor, these data suggest that increased folding in ACBI2 led to a reduction in the efflux ratio, contributing to improved oral bioavailability (Extended Data Table **4**).

5) Page 11, line 46. “These degree” should read “The degree”.

Thank you to the reviewer for spotting this error – we have corrected it.

6) The headings of the Extended Data Tables are placed after the tables. It would be better if they are placed at the top of the tables. In addition, it is difficult for the non-specialist to understand what

data is given in the columns of the tables. For instance, I struggled with Extended Data Table 1; e.g. what is the VCB+SMARCA2 KD in column 4 as compared to the SMARCA2 KD given in column 10. Considering that such a large volume of important data is presented in the Extended Data I strongly recommend that the authors spend time inserting footnotes that clearly describe what data is found in the tables.

Thank you to the reviewer for pointing this out. As requested by the editor, the Extended Data Tables have been reformatted and removed from the main manuscript. They can now be found as Supplementary Data 1-5, also referenced like this in the main manuscript, as individual Supplementary Data Files. The legend for these files is provided in a separate word document. The legend for Supplementary Data 1 is rephrased to bring more clarity to the reader as follows:

Supplementary Data 1: Measurement of binding affinities towards SMARCA2BD and SMARCA4BD \pm \pm VHL-ElonginC-ElonginB complex (VCB), cooperativities (α) and ternary complex half-lives via SPR, errors are \pm standard deviation with repeats (n) specified in brackets. (α) is calculated as ratio between SMARCA KD / SMARCA KD +VCB values measured in the same run, respectively. Source data are provided as a Source Data file.